# Are Population Graphs Really as Powerful as Believed?

Tamara T. Mueller[1]    Sophie Starck[1]    Kyriaki-Margarita Bintsi[2]
Alexander Ziller[1]    Rickmer Braren[3]    Georgios Kaissis[1,3,4]    Daniel Rueckert[1,2]

[1] *AI in Medicine and Healthcare, Technical University of Munich, Germany*
[2] *BioMedIA, Imperial College London, UK*
[3] *Institute for Diagnostic and Interventional Radiology, Technical University of Munich, Germany*
[4] *Machine Learning in Biomedical Imaging, Helmholtz Munich, Germany*
*Contact: tamara.mueller@tum.de*

**Reviewed on OpenReview:** *https://openreview.net/forum?id=TTRDCVnbjI*

## Abstract

Population graphs and their use in combination with graph neural networks (GNNs) have demonstrated promising results for multi-modal medical data integration and improving disease diagnosis and prognosis. Several different methods for constructing these graphs and advanced graph learning techniques have been established to maximise the predictive power of GNNs on population graphs. However, in this work, we raise the question of whether existing methods are really strong enough by showing that simple baseline methods –such as random forests or linear regressions–, perform on par with advanced graph learning models on several population graph datasets for a variety of different clinical applications. We use the commonly used public population graph datasets TADPOLE and ABIDE, a brain age estimation and a cardiac dataset from the UK Biobank, and a real-world in-house COVID dataset. We (a) investigate the impact of different graph construction methods, graph convolutions, and dataset size and complexity on GNN performance and (b) discuss the utility of GNNs for multi-modal data integration in the context of population graphs. Based on our results, we argue towards the need for "better" graph construction methods or innovative applications for population graphs to render them beneficial[1].

## 1 Introduction

Graphs can be used to model and represent various types of data. They allow for a suitable representation of interconnected structures, such as social networks (Fan et al., 2019), molecules (Moreira-Filho et al., 2022), or surface meshes (Mueller et al., 2023b). In order to perform deep learning on graph-like data structures, graph neural networks (GNNs) have been introduced (Gori et al., 2005; Scarselli et al., 2008). GNNs follow a message-passing scheme and collect information that is stored in nodes across a graph structure (Bronstein et al., 2017) and have shown improved performance of various deep learning tasks (Parisot et al., 2017; Ahmedt-Aristizabal et al., 2021; Bessadok et al., 2022; Pellegrini et al., 2022). Most of these tasks rely on datasets that inherently provide a graph structure, such as social networks, or provide well-established methods to construct the graph, such as point clouds (Wang et al., 2019).

In the medical domain, GNNs have been applied to improve disease diagnostics (Parisot et al., 2017; Cosmo et al., 2020; Kazi et al., 2022), model biological structures (Chen et al., 2020), or temporal components of data (Kim et al., 2021). They can be used to perform deep learning on surface meshes for fatty tissue quantification (Mueller et al., 2023b), vessel structures (Paetzold et al., 2021) for vessel segmentation, or molecules for drug discovery (Bonner et al., 2022). The respective datasets provide an inherent graph structure in the form of a mesh, a vessel tree, or chemical bindings. In contrast to datasets that provide a clear graph structure,

---

[1] The source code for this work can be found at: **https://github.com/tamaramueller/population_graphs**

one research area in medicine studies so-called *population graphs*. A population graph refers to a network of inter-connected subjects encoding the medical information of all subjects in graph form. Usually, the subjects' medical data, such as imaging or clinical features, is used as node features in the graph. The edges are constructed so that similar subjects are connected. Figure 1 shows a schematic of a typical population graph. Each subject (node) is represented by a data vector often extracted from medical images. Additionally, non-imaging clinical data, such as demographics or lab results, can be used to define the edges between subjects, where similar non-imaging features lead to a connection between two subjects.

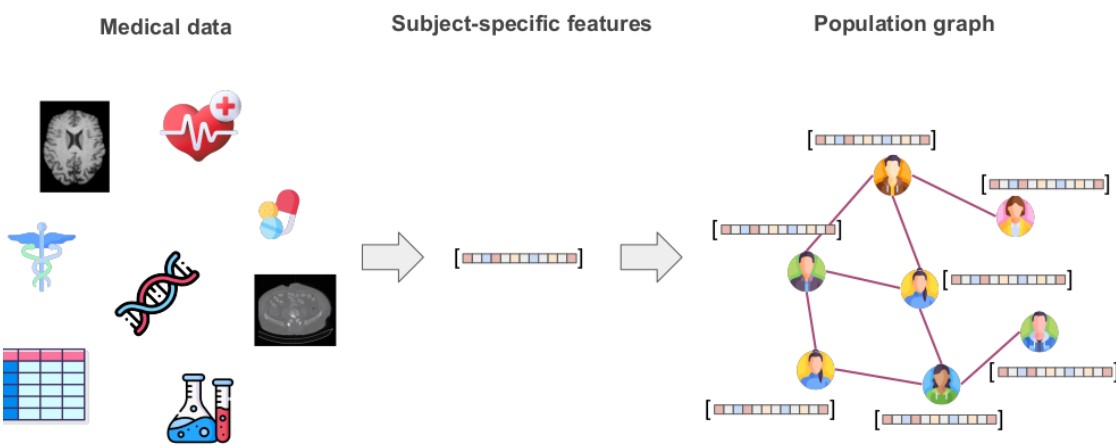

Figure 1: **Overview of a typical population graph construction**. Subject-specific medical data is represented as a feature vector and used as node features in the population graph. The most frequently used setup uses imaging features as node features and non-imaging features for edge construction.

Several works have shown that population graphs for medical applications can improve downstream tasks compared to graph-agnostic methods (Parisot et al., 2017; Kazi et al., 2019; Cosmo et al., 2020; Kazi et al., 2022). Parisot et al. (2017) first introduced the concept of population graphs for the detection of Alzheimer's disease and autism. Later works (Kazi et al., 2019; Cosmo et al., 2020; Kazi et al., 2022; Bintsi et al., 2023a;b) used the method of population graphs under different settings, developing new graph construction methods and for different tasks, such as age prediction. The motivation for using population graphs is the hypothesis that subjects that share similar phenotypes tend to have similar pathologies and, therefore, benefit from sharing information. The goal is to facilitate personalised medicine by utilising the shared information across similar subjects. However, population graphs come with a significant limitation: the graph structure needs to be constructed from the dataset. This has led to different graph construction methods. Two branches of graph construction have been established: static and dynamic graph construction. Static graph construction refers to creating the graph structure prior to graph learning, while dynamic graph construction methods adapt the graph structure during training (Cosmo et al., 2020). To date, both methods are used frequently. For an overview of graph construction methods for GNNs in medicine, we refer to Mueller et al. (2024). What makes the choice of graph construction method so crucial is the impact of the resulting graph structure on the downstream performance of the GNN. It has been shown that a "poor" graph structure can lead to GNNs under-performing graph-agnostic models (Luan et al., 2022; Zhu et al., 2020). Some methods have been specifically designed to work on such challenging graph structures, one of them being neural sheaf diffusion models (Hansen & Gebhart, 2020). We investigate their potential on population graph datasets, which tend to have challenging graph structures.

So far, there are two commonly used arguments for using medical population graphs compared to graph-agnostic models: (1) GNNs allow for meaningful multi-modal data integration, and (2) the message passing across neighbourhoods improves model performance. In this work, we investigate how firm those claims are and contradict them on several datasets. Our contributions can be summarised as follows:

- We compare static and dynamic state-of-the-art graph construction methods with GNNs, as well as the usage of neural sheaf diffusion models for population graphs and show how simple graph-agnostic baselines perform on par with them on several population graph datasets.

- We show that GNNs can be superior to graph-agnostic models if the graph structure is provided with the dataset but do not achieve performance boosts on any medical population graph dataset used in this work. We hypothesise that in the latter case, the graph structure does not add additional valuable information.

- We evaluate the impact of the graph structure on several different types of graph convolution using two different graph assessment metrics: homophily and cross-class neighbourhood similarity (CCNS) distance.

- We highlight that the graph construction methods for population graphs are not sufficient and discuss potential future directions for population graph studies.

Our results lead us to conclude that we need a discussion about whether population graphs are beneficial over graph-agnostic methods and that the currently available graph construction methods are the performance bottleneck of GNNs on population graphs. We see a requirement for "better" graph construction methods if we want to improve the performance of GNNs on population graphs.

## 2 Background

In this section, we discuss some background on graphs, graph neural networks, neural sheaf diffusion models, and two graph assessment metrics, namely homophily and cross-class neighbourhood similarity.

### 2.1 Graph Structures

A graph $G := (V, E)$ is defined as a set of $n$ vertices/nodes $V$ and a set of edges $E$, where $e_{ij} = 1$ and $e_{ij} \in E$ if there exists an edge from node $i$ to node $j$. All edges can be summarised in an $n \times n$ adjacency matrix $\mathbf{A}$, where $a_{ij} = 1$ if $e_{ij} \in E$ and 0 otherwise. In the context of graph deep learning, the graph's nodes usually hold node features of dimension $r$ that can be summarised in the node feature matrix $\mathbf{X} \in \mathbb{R}^{n \times r}$. A neighbourhood of a node $i$, $\mathcal{N}_i$ is the set of all nodes $j$, for which an edge $e_{ji}$ from $j$ to $i$ exists. Furthermore, in the setting of node classification, each node $i$ usually holds a label $y_i$, and all labels can be summarised in the label vector $Y$.

### 2.2 Graph Assessment Metrics

Several works have shown that the graph structure can have a significant impact on the performance of GNNs (Luan et al., 2022; Zhu et al., 2020). In this line, different metrics have been introduced that assess graph structures and have been shown to correlate with GNN performance. The metric most commonly used is *homophily*. One can distinguish between three different types of homophily: class homophily (Lim et al., 2021; Luan et al., 2021), edge homophily (Kim & Oh, 2022), and node homophily (Pei et al., 2020), which all highlight slightly different aspects of the graph structure. They all evaluate the ratio between edges that connect nodes with the same label and edges that connect nodes with different labels. The idea is that since GNNs propagate node features across edges, the less similar the neighbours are, the less likely it is for the GNN to learn representative node feature embeddings for this node, which can impact the network's performance. In the remaining parts of this work, we will use node homophily.

**Definition 2.1 (Node homophily (Pei et al., 2020))** *A graph $G := (V, E)$ with node labels $Y := \{y_u; u \in V\}$ has the following node homophily:*

$$\mathrm{hom}(G, Y) := \frac{1}{|V|} \sum_{v \in V} \frac{|\{u | u \in \mathcal{N}_v, Y_u = Y_v\}|}{|\mathcal{N}_v|}, \tag{1}$$

*where $\mathcal{N}_v$ is the set of neighbouring nodes of $v$ and $|\cdot|$ the cardinality of a set.*

We speak of "high homophily" or a "homophilic" graph, when $\mathrm{hom}(G, Y) \to 1$ and of "low homophily" or a "heterophilic" graph, when $\mathrm{hom}(G, Y) \to 0$. A graph's homophily can also be defined for regression tasks by taking the distance between node feature labels among neighbourhoods into account (Mueller et al., 2023a):

**Definition 2.2 (Homophily for regression (Mueller et al., 2023a))** *The node homophily of a graph $G$ with labels $Y$ (defined as above) that indicate a regression task is defined as follows:*

$$\mathrm{hom}_{\mathrm{reg}}(G, Y) := 1 - \left( \frac{1}{|V|} \sum_{v \in V} \left( \frac{1}{|\mathcal{N}_v|} \sum_{n \in \mathcal{N}_v} \|y_v - y_n\|_1 \right) \right), \tag{2}$$

*where $\|\cdot\|_1$ indicates the $L_1$ norm.*

Another metric that does not only focus on the ratio of edges connecting same-labelled or differently-labelled nodes is cross-class neighbourhood similarity (CCNS) (Ma et al., 2021). Here, the overall similarity of neighbourhoods of nodes with the same label is evaluated, irrespective of whether the neighbours share the same label as the node of interest.

**Definition 2.3 (Cross-class neighbourhood similarity (Ma et al., 2021))** *Let $G := (V, E)$, $Y$, and $\mathcal{N}_v$ be defined as above. In addition, let $C$ be the set of all possible classes of node labels, and $V_c$ the set of vertices of a specific class $c$. Then the CCNS of two classes $c_r$ and $c_s$ can be derived as follows:*

$$\mathrm{CCNS}(c_r, c_s) = \frac{1}{|V_{c_r}||V_{c_s}|} \sum_{u,v \in V} \mathrm{cossim}(d(u), d(v)), \tag{3}$$

*where $d(v)$ indicates the histogram of a node $v$'s neighbours' labels and $\mathrm{cossim}(\cdot, \cdot)$ the cosine similarity.*

Mueller et al. (2023a) introduce a reduction of CCNS to a single-valued parameter, they call *CCNS distance*, which defines the $L_1$ distance between the CCNS matrix and the identity matrix:

**Definition 2.4 (CCNS distance (Mueller et al., 2023a))** *Let $G := (V, E)$, $C$, and $\mathrm{CCNS}$ be defined as above. Then the CCNS distance of the whole graph $G$ is:*

$$D_{\mathrm{CCNS}} := \frac{1}{n} \sum \|\mathrm{CCNS} - \mathbb{I}\|_1, \tag{4}$$

*where $\|\cdot\|_1$ is the $L_1$ norm and $\mathbb{I}$ the identity matrix.*

### 2.3 Graph Neural Networks

GNNs have been introduced with the aim of enabling deep learning on non-Euclidean spaces, such as graphs, manifolds, or meshes (Bronstein et al., 2017). They all follow a so-called message-passing scheme, which propagates the information that is stored in the node features of the graph (or mesh or manifold) to its neighbouring nodes. The GNN then learns a node feature embedding based on the original node features as well as the propagated node features of the neighbouring nodes. GNNs make use of graph convolutions, which specify the concrete message-passing scheme that is applied during training and inference. There exist several different types of graph convolution, all varying slightly in their methodology. Here, we summarise the definitions of four commonly used graph convolutions.

**Definition 2.5 (Graph Convolutional Networks (GCN) (Kipf & Welling, 2016))** *Graph convolutional networks (GCNs) were one of the first GNNs introduced by Kipf & Welling (2016). They were originally defined in a spectral manner, using the graph Laplacian. The `PyTorch Geometric` implementation follows the following definition:*

$$x_i' = \Theta^T \sum_{j \in \mathcal{N}_i \cup \{i\}} \frac{1}{\sqrt{\hat{d}_j \hat{d}_i}} \, x_j, \tag{5}$$

*where $\hat{d}_i = 1 + \sum_{j \in \mathcal{N}_i} 1$, $\Theta$ learnable weights, and $\mathcal{N}_i$ the neighbourhood of node $i$.*

**Definition 2.6 (Graph SAGE (Hamilton et al., 2017))** *In 2017, Hamilton et al. (2017) introduced a novel graph convolution that was originally designed for large graphs and inductive training, which is called GraphSAGE. Here, the new feature representation of a node i is defined as follows:*

$$x_i' = W_1 x_i + W_2 \cdot \mathbb{E}_{j \in \mathcal{N}_i}, \tag{6}$$

*where $W_1$ and $W_2$ denote learnable weights and $\mathbb{E}_{j \in \mathcal{N}_i}$ the expectation over all node features in the neighbourhood of $j$.*

**Definition 2.7 (Higher-order Graph Neural Networks (GraphCONV) (Morris et al., 2019))** *Morris et al. (2019) introduced so-called higher-order GNNs, where the node feature embedding $x_i'$ of node i is defined as follows:*

$$x_i' = W_1 x_i + W_2 \sum_{j \in \mathcal{N}_i} x_j, \tag{7}$$

*where $W_1$ and $W_2$ are learable weights and $\mathcal{N}_i$ denotes the neighbourhood of node $i$.*

**Definition 2.8 (Graph Attention Networks (GAT) (Veličković et al., 2017))** *Veličković et al. (2017) introduced a graph neural network, that learns attention weights for edges in the graph. The new node feature embedding of a node i is defined as:*

$$x_i' = \alpha_{ii} \Theta x_i + \sum_{j \in \mathcal{N}_i} \alpha_{ij} \Theta x_j, \tag{8}$$

*where $\Theta$ are learable parameters and $\alpha_{ij}$ is the attention coefficient between two nodes i and j and is defined as follows:*

$$\alpha_{ij} = \frac{\exp\left(\phi\left(a^T\left(\Theta x_i \parallel \Theta x_j\right)\right)\right)}{\sum_{k \in \mathcal{N}_i \cup i} \exp\left(\phi\left(a^T\left(\Theta x_i \parallel \Theta x_k\right)\right)\right)}, \tag{9}$$

*where $\phi$ is commonly the `LeakyReLU` function and $\parallel$ indicates a concatenation of the values.*

## 2.4 Neural Sheaf Diffusion Models

With a rising discussion on how GNNs perform on low-homophily graph structures, different approaches to graph learning have been established that target these more challenging settings for graph learning. One of these methods is neural sheaf diffusion models, originally introduced by Hansen & Gebhart (2020) and extended by Bodnar et al. (2022). They use the topological concept of cellular sheaves, which assign vector spaces to all nodes and edges and linear mappings between them for all node-edge connections. Traditional GNNs are designed in a way that they assume a graph structure with a trivial underlying sheaf. Hansen & Gebhart (2020) and Bodnar et al. (2022) introduce an alternative approach to graph deep learning that is based on the concept of cellular sheaves, where different sheaf representations are learned for nodes and edges of the graph. They show that with this method, they can provide a graph learning technique that is less impacted by heterophilic graphs and over-smoothing - two commonly known limitations of GNNs. Sheaf neural networks (Hansen & Gebhart, 2020; Bodnar et al., 2022) are a generalisation of GCNs (Kipf & Welling, 2016) and leverage the sheaf Laplacian (Hansen & Ghrist, 2019), an extension of the graph Laplacian. This allows for an expression of more complex relationships between nodes rather than "similarity". Bodnar et al. (2022) furthermore show how these sheaves can be learned from the data at hand, using neural networks.

**Definition 2.9 (Sheaf Convolution)** *Let $\mathcal{F}$ be a sheaf on a graph G with feature matrix $X \in \mathbb{R}^{nd \times a}$ and sheaf Lapacian $\Delta_{\mathcal{F}}$. A sheaf convolutional model is then defined as follows:*

$$Y = \sigma\left(\left(I_{nd} - \Delta_{\mathcal{F}}\right)\left(I_n \otimes W_1\right) X W_2\right), \tag{10}$$

*where $\sigma$ is a non-linearity, $\otimes$ denotes the Kronecker product, $W_1 \in \mathbb{R}^{d \times d}$ and $W_2 \in \mathbb{R}^{a \times b}$ are two weight matrices, and a and b define the number of input and output channels, respectively.*

The authors introduce different versions of neural sheaf networks, such as *GeneralSheaf*, *BundleSheaf*, and *DiagSheaf*. For more details about sheaf networks, we refer to Hansen & Gebhart (2020) and Bodnar et al. (2022). In this work, we utilise neural sheaf diffusion models on all classification datasets in order to investigate their potential on potentially low-homophily graph structures of medical population graphs.

# 3 Related Work

Medical population graphs have been used for several different downstream tasks, such as disease prediction (Parisot et al., 2017; Kazi et al., 2019; 2022) or age prediction (Bintsi et al., 2023a;b). Given that the construction of the graph itself is a major challenge when working with population graphs, several methods for graph construction have been established, which we utilise and compare in this work. For example, dynamic graph learning (Cosmo et al., 2020; Kazi et al., 2022) has been established to allow for end-to-end learning of the graph structure, so the graph does not have to be defined manually. There is little work investigating the impact of different graph construction methods and different graph learning schemes on the performance of population graphs. Bintsi et al. (2023b), for instance, evaluate different static graph construction methods on an age regression dataset but do not evaluate dynamic graph construction methods. To the best of our knowledge, this is the first work specifically addressing the challenge of graph construction in population graph studies in combination with different graph learning methods and with a detailed comparison to baseline models.

In general GNN research, several works have investigated the impact of the graph structure on model performance. Zhu et al. (2020) address the issue of the impact of the graph structure, measured by homophily (see Section 2.2), on different graph convolutional networks on citation networks. Several metrics have been established that allow for an assessment of the graph structure and show a correlation with the performance of GNNs. Luan et al. (2022) introduce two metrics, normalised total variation and normalised smoothness value, that measure the effect of edge bias. Xie et al. (2020) measure the graph structure with two metrics called neighbourhood entropy and centre-neighbourhood similarity. Ma et al. (2021) utilise the above-mentioned metric called cross-class neighbourhood similarity, which assesses how similar all neighbourhoods of all nodes with the same label are and show their correlation with GNN performance. Most of these works assess their metrics on benchmark datasets, such as citation networks, that come with a ground truth graph structure. In this work, we want to take these experiments one step further and investigate the impact of graph construction methods on population graph studies with GNNs and investigate the benefit of using GNNs over baseline methods.

# 4 Methods and Training Setup

In this section, we provide an overview of the utilised methods in this work. We introduce the different static and dynamic graph construction methods, summarise the utilised GNN models and the training setup, and introduce the datasets that were used to perform the experiments. A summary of the different learning and graph construction pipelines is visualised in Figure 2.

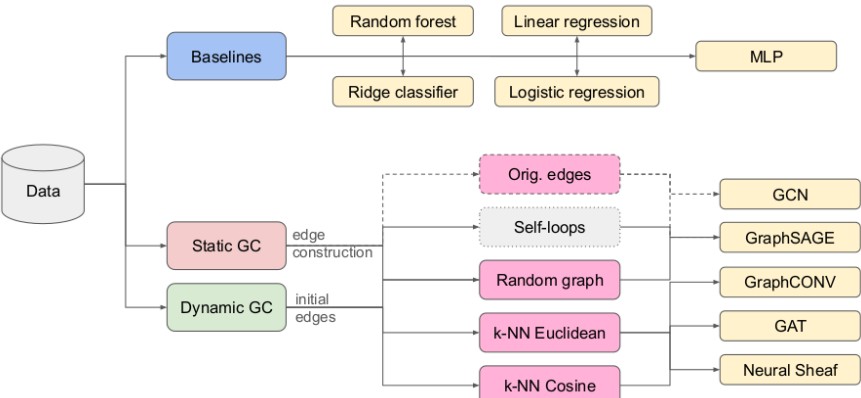

Figure 2: **Overview of the conducted experiments.** We tune different baselines and compare their performance to GNNs on population graphs. We perform static and dynamic graph construction (GC) and use four graph convolutions: GCN, GraphSAGE, GraphConv, and GAT, and Neural Sheaf Models. The original edges are only used if available, and self-loops mimic a transductive learning setting (see appendix).

## 4.1 Datasets

We perform our experiments on five medical population graph datasets, which are summarised in Table 1. First, we use the commonly used subset of the **TADPOLE** dataset (Yu et al., 2020) that is, for example, used in Kazi et al. (2022). The task of this dataset is to distinguish between patients with Alzheimer's disease (AD), ones with mild cognitive impairment (MCI), and healthy control groups (NC). The dataset consists of 30 imaging features of 564 subjects. A second public and frequently used dataset for population graph studies is the Autism Brain Imaging Data Exchange (**ABIDE**) dataset (Di Martino et al., 2014). It contains brain imaging features and clinical features such as age of 871 subjects and has been used in the context of population graphs in several works (Parisot et al., 2017; Kazi et al., 2019; 2022). The task of this dataset is a binary classification task, discriminating between autism patients and healthy controls. Furthermore, we use a small real-world medical dataset of **COVID** patients that has also been used before in population graph settings (Keicher et al., 2021); however, in a slightly different version of the dataset. The task is a binary classification of whether a subject is predicted to require intensive care or not. The dataset consists of image-derived features and clinical features of 65 subjects. Additionally, we use a larger population graph dataset from the UK Biobank (UKBB) (Sudlow et al., 2015) that consists of features extracted from brain magnetic resonance (MR) images (**UKBB brain age**). To extract the features, we follow the approach from Cole (2020), resulting in 68 imaging features and 20 non-imaging features for each subject. We use a set of 6406 subjects and perform a regression task for age prediction on this dataset. The mean age of this dataset is 62.86 years. We use this dataset to explore the difference in model performance when only using the imaging features compared to using all features. If not specifically specified, we only use the 68 imaging features. We extract another dataset from the UKBB (Sudlow et al., 2015) containing imaging features from cardiac MRIs as well as clinical features, on which we perform a binary classification of whether a subject suffers from cardiovascular diseases or not (**UKBB cardiac**). We extract 6 non-imaging features and 86 imaging features using the pipeline from Bai et al. (2020) and create a population graph with 2900 subjects.

Table 1: **Overview of all utilised population graph datasets** with the respective number of nodes, number of samples/nodes in the train, test, and validation sets, the number of node features (Nr. features), and the number of classes.

| Dataset | Nr. nodes | Train samples | Val. samples | Test samples | Nr. features | Nr. classes |
|---|---|---|---|---|---|---|
| TADPOLE | 564 | 468 | 48 | 57 | 30 | 3 |
| ABIDE | 871 | 609 | 41 | 221 | 6105 | 2 |
| UKBB cardiac | 2900 | 2320 | 58 | 522 | 89 | 2 |
| COVID | 65 | 45 | 4 | 16 | 29 | 2 |
| UKBB brain age | 6406 | 4811 | 1276 | 319 | 88 | Regression |

In order to evaluate the impact of the graph construction method and the resulting graph structure on the performance of the GNN, we also utilise three benchmark citation datasets: **CORA**, **CITESEER**, and **PUBMED** (Yang et al., 2016). These datasets come with a pre-defined graph structure, which we can use as the ground truth graph and compare performance to our generated graph structures. In Section 5.4, we also evaluate the impact of scale with a **synthetically** generated classification dataset with 4 classes and between 5 000 and 30 000 nodes.

## 4.2 Graph Construction Methods

We use distinct graph construction methods for population graphs and compare their impact on the performance of different GNNs. We note that the utilised methods are not extensive, but we picked the most representative, most frequently used, and well-established methods for static and dynamic graph construction. For more details on graph construction methods for GNNs in medicine, we refer to Mueller et al. (2024).

### 4.2.1 Static Graph Construction

Static graph construction methods refer to the construction of a graph structure that stays constant throughout GNN training. There are several methods to construct a static population graph structure, while the most common one utilises a $k$-nearest neighbour approach (Cunningham & Delany, 2021).

**Self-loops Only**  To get an intuition about the impact of the graph structure on the GNN, we evaluate a GNN on a graph that is not really a graph but only contains self-loops. The adjacency matrix of a graph that only contains self-loops is equivalent to the identity matrix. In this setting, no message passing among nodes is performed since there are no connections between nodes. We use this setting to simulate a transductive learning setting without using a graph structure.

**Random Graph**  Secondly, we construct a random graph structure by generating an Erdos-Rényi Graph with an edge probability of 0.001. We choose to evaluate all methods applied to a graph with a random graph structure in order to investigate the impact of the graph structure on model performance.

**$k$-Nearest Neighbour Graph**  The most frequently used approach of graph construction for population graphs is the $k$-Nearest Neighbour ($k$-NN) approach. Here, $k$ is a hyperparameter and defines the number of neighbours each node has. For this approach, different distance measures can be used, for example, the Euclidean distance or the cosine similarity. We use the implementation of `knn_graph` from Pytorch Geometric (Fey & Lenssen, 2019) and refer to the usage of the Euclidean distance as "$k$-NN Eucl." and the usage of the cosine similarity as "$k$-NN Cosine" in the tables below.

### 4.2.2 Dynamic Graph Construction

Dynamic graph construction refers to the learning of the graph structure in an end-to-end manner in parallel to the model training. There exist a few dynamic graph construction methods; however, for population graphs, mostly the approach from Kazi et al. (2022) is used. Here, we use the dDGM method, a differentiable graph construction method that allows for end-to-end learning of the graph structure during GNN training. In their work, Kazi et al. (2022) propose two differentiable graph learning modules: cDGM and dDGM. We here only use the dDGM implementation since both in their work and in our preliminary results and related works like (Mueller et al., 2023a), dDGM resulted in better performance. The dDGM module can be applied to arbitrary initial graph structures. We evaluate the impact of the initial graph structure on the model performance by using different graphs as a starting point. For the CORA dataset, we evaluate dDGM starting with **(a)** no edges, **(b)** only self-loops, **(c)** a random graph structure, **(d)** a $k$-NN graph, and **(e)** the original edges of the dataset, in Section 5.4.

### 4.3 Graph Assessment

In order to gain insights into the constructed graph structures and investigate their "quality", we evaluate two graph assessment metrics: node homophily (Pei et al., 2020) and cross-class neighbourhood similarity (CCNS) (Ma et al., 2021). We follow the approach from Mueller et al. (2023a) and evaluate the *CCNS distance*, the there-defined homophily for regression tasks, and split the evaluation of all metrics into train and test nodes. The latter can be useful to investigate how differently the graph structure impacts training and test nodes.

### 4.4 Model Architectures and Training

We use two different model architectures in our experiments. For all dynamic graph construction experiments, we use the architecture proposed by Kazi et al. (2022), which consists of two graph convolutional networks: a graph embedding function $f$ and a diffusion function $g$. Following the results from the original paper (Kazi et al., 2022), we use the respective graph convolutions for both modules. For the static graph construction experiments, we use a GNN with 1, 2, or 3 graph convolutional layers (e.g. GCN or GraphSAGE), followed by an MLP. We use two sets of hyperparameters regarding the layers of these networks that can be found in the Appendix. During preliminary experiments, we noticed that using the same architecture for static graph construction results in strong over-fitting of the models to the training sets. We, therefore, use a different

architecture for the static graph construction experiments than for the dynamic ones. More details about all architectures can be found in the appendix. In all architectures, we utilise four different frequently used graph convolutions, namely graph convolutional networks (GCNs) (Kipf & Welling, 2016), graph SAGE networks (Hamilton et al., 2017), higher-order GNNs (GraphConv) (Morris et al., 2019), and graph attention networks (GATs) (Veličković et al., 2017). They all differ in the methodology of how the message-passing scheme is performed and their formal definitions can be found in Section 2.3. For the neural sheaf diffusion models, we utilise the setup of the original work, varying between the following sheaf models: *BundleSheaf*, *DiagSheaf*, and *GeneralSheaf*.

All models are trained in a transductive setting, where all nodes are available during training. We define a fixed set of hyperparameters for all experiments and run a hyperparameter search for at least 200 runs using sweeps from *Weights and Biases* (Biewald, 2020). We then pick the run with the best validation accuracy/MAE, evaluate its performance over 5 random seeds, and report the mean test accuracy with the standard deviation. All trainings are performed on an Nvidia Quadro RTX 8000 GPU, using `Pytorch lightning` and `Pytorch Geometric` (Fey & Lenssen, 2019). The hyperparameters can be found in the appendix.

## 5 Experiments and Results

In this section, we summarise our experiments with different graph construction methods, including static and dynamic graph construction and Neural Sheaf Diffusion models. We (1) summarise the overall best-performing GNNs for all datasets and compare them to three different baselines and discuss more detailed results on two of the medical population graph datasets, (2) compare our results to different state-of-the-art (SOTA) population graph studies, (3) evaluate the method of population graphs for multi-modal data integration, and (4) evaluate the impact of the different components –such as graph structure, dataset complexity and size– on the performance of GNNs for population graphs.

The most noteworthy finding of our work is possibly the fact that simple baseline methods outperform more complex graph learning techniques on all tested population graph datasets.

### 5.1 Baselines Achieving Comparable Performance to GNNs

During an extensive evaluation of the performance of GNNs on medical population graphs, we found that when optimally tuning baseline models (random forest, linear/logistic regression, and ridge classifier/regression) they perform competitively on all datasets. We summarise these results, the best GNN as well as a Neural Sheaf Diffusion model in Table 2, where the best model for each dataset is highlighted in bold.

Table 2: **Summary of results** of different baseline methods and the best GNNs and Neural Sheaf Models, either from our training evaluated on 5 random seeds or from literature ([1]: Parisot et al. (2017)). For classification datasets, we report the test accuracy; for regression tasks, the test MAE.

| Method | TADPOLE | UKBB Brain Age | UKBB Cardiac | COVID | ABIDE |
|---|---|---|---|---|---|
| Random forest | $0.9474 \pm 0.00$ | $3.7913 \pm 0.01$ | $0.7061 \pm 0.01$ | $0.8250 \pm 0.02$ | $0.7046 \pm 0.01$ |
| Ridge | $0.7368 \pm 0.00$ | $3.4185 \pm 0.00$ | $0.6935 \pm 0.00$ | $0.8750 \pm 0.00$ | $0.7014 \pm 0.00$ |
| Linear/Logistic | $0.8421 \pm 0.00$ | $3.4287 \pm 0.00$ | $0.6858 \pm 0.00$ | $0.8125 \pm 0.00$ | $0.6290 \pm 0.00$ |
| GNN $k$-NN | $0.9404 \pm 0.02$ | $3.3524 \pm 0.06$ | $0.6970 \pm 0.02$ | $0.7875 \pm 0.03$ | $0.695$ [1] |
| Neural Sheaf | $0.9368 \pm 0.02$ | - | $0.6904 \pm 0.01$ | $0.8000 \pm 0.03$ | $0.5448 \pm 0.01$ |

It is noteworthy that for all population graph datasets apart from the UKBB brain age dataset, at least one of the baseline methods outperforms the best GNN model. On the UKBB brain age dataset, the GNN slightly outperforms the ridge regression (best baseline) by an MAE of 0.066. However, a two-sided $t$-test between the results of the best GNN and the strongest baseline (ridge regression) did not show a significant difference in performance with a $p$-value of 0.06. These results raise the main question of this work: *"Are population graphs really as powerful as believed?"* Our results indicate the contrary, and we investigate the discrepancy between our work and related works in the following sections, discussing potential reasons for this gap.

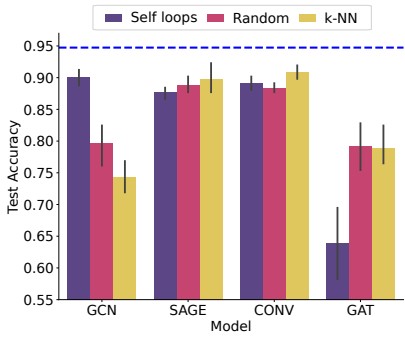

(a) **Static** graph construction on **TADPOLE**

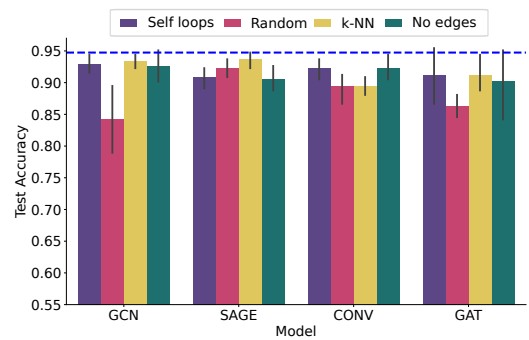

(b) **Dynamic** graph construction on **TADPOLE**

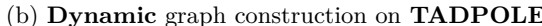
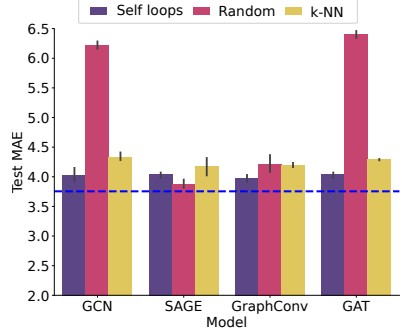

(c) **Static** graph construction on **UKBB brain age**

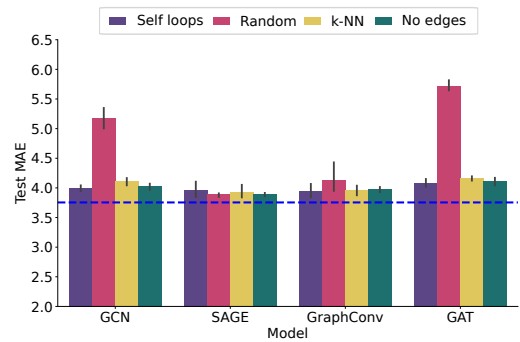

(d) **Dynamic** graph construction on **UKBB brain age**

Figure 3: **Results on two datasets** with static graph construction (left column) and dynamic graph construction (right column). First row: **TADPOLE** reporting the test accuracy (higher better), second row: **UKBB brain age**, reporting the test MAE (lower better). The mean performance of the baseline is indicated by the dashed blue lines.

In the following, we evaluate the experiments summarised in Figure 2 on the two population graph datasets TADPOLE and UKBB brain age in more detail. The results are visualised in Figure 3, where the first row shows the TADPOLE dataset and the second row the UKBB brain age dataset. The results are also listed in Tables 3 and 4, respectively. For the TADPOLE dataset, none of the GNNs outperform the best baseline method, which in this case is a random forest. This is even the case in settings where the homophily of the test set is very high, for example, for the static $k$-NN graph construction and the GAT convolution. We observe similar results on the UKBB brain age dataset, where we perform age regression on the imaging features only and report the MAE as model performances. We do not report the CCNS values for this dataset since CCNS is not defined for regression tasks. GraphSAGE and GraphConv networks do not seem to be influenced by the randomness of the graph structure and are still able to learn meaningful representations of the node features and make accurate predictions. The homophily of the $k$-NN graphs generated for the UKBB dataset is also quite high, similar to the TADPOLE dataset. The same holds for its low CCNS distance score. Furthermore, we observe that GCN models tend to perform better at a lower number of neighbours. Interestingly, the best-performing GNN on the TADPOLE dataset is trained on a random graph structure, using GraphSAGE convolutions and dynamic graph construction. We also cannot see a clear benefit of using dynamic graph construction methods on all datasets. While the best dynamic result outperforms the best static result on the TADPOLE dataset, static methods achieve higher results on the UKBB brain age dataset. We observe the same behaviours on all other datasets Their results can be found in the appendix.

Table 3: Results of the experiments on the **TADPOLE** dataset. GC: graph construction, BL: baselines, $k$: number of neighbours. The best performance for each method is bold.

| | Initial edges | Model | $k$ | Test acc ↑ | Test homophily ↑ | Test CCNS distance ↓ |
|---|---|---|---|---|---|---|
| **BL** | - | Majority vote | - | $0.5674 \pm 0.00$ | - | - |
| | - | Random forest | - | $\mathbf{0.9474 \pm 0.00}$ | - | - |
| | - | Logistic regression | - | $0.8597 \pm 0.00$ | - | - |
| **Static GC** | Random | GCN | - | $0.7965 \pm 0.04$ | $0.426 \pm 0.49$ | 0.348 |
| | | SAGE | - | $0.8877 \pm 0.01$ | $0.426 \pm 0.49$ | 0.348 |
| | | GraphConv | - | $0.8842 \pm 0.01$ | $0.426 \pm 0.49$ | 0.348 |
| | | GAT | - | $0.7930 \pm 0.04$ | $0.426 \pm 0.49$ | 0.348 |
| | $k$-NN Euclidean | GCN | 5 | $0.7439 \pm 0.03$ | $0.775 \pm 0.24$ | 0.213 |
| | | SAGE | 5 | $0.8982 \pm 0.03$ | $0.775 \pm 0.24$ | 0.213 |
| | | GraphConv | 5 | $\mathbf{0.9088 \pm 0.01}$ | $0.775 \pm 0.24$ | 0.213 |
| | | GAT | 2 | $0.7895 \pm 0.04$ | $\mathbf{0.904 \pm 0.20}$ | **0.094** |
| **Dynamic GC** | No edges | GCN | 20 | $0.9263 \pm 0.03$ | $\mathbf{0.919 \pm 0.19}$ | 0.073 |
| | | SAGE | 20 | $0.9053 \pm 0.02$ | $0.806 \pm 0.21$ | 0.183 |
| | | GraphConv | 2 | $0.9228 \pm 0.02$ | $0.798 \pm 0.34$ | 0.190 |
| | | GAT | 20 | $0.9018 \pm 0.06$ | $0.908 \pm 0.15$ | **0.101** |
| | Random | GCN | 2 | $0.8421 \pm 0.06$ | $0.851 \pm 0.27$ | 0.177 |
| | | SAGE | 10 | $0.9228 \pm 0.02$ | $0.423 \pm 0.22$ | 0.616 |
| | | GraphConv | 5 | $0.8947 \pm 0.03$ | $0.411 \pm 0.25$ | 0.594 |
| | | GAT | 5 | $0.8632 \pm 0.02$ | $0.895 \pm 0.20$ | 0.119 |
| | $k$-NN Euclidean | GCN | 2 | $0.9333 \pm 0.01$ | $0.793 \pm 0.28$ | 0.204 |
| | | SAGE | 20 | $\mathbf{0.9368 \pm 0.01}$ | $0.461 \pm 0.63$ | 0.632 |
| | | GraphConv | 10 | $0.8947 \pm 0.02$ | $0.777 \pm 0.29$ | 0.219 |
| | | GAT | 10 | $0.9123 \pm 0.03$ | $0.775 \pm 0.29$ | 0.206 |

Table 4: Results of the experiments on the **UKBB brain age** imaging dataset. BL: baselines, $k$: number of neighbours, GC: graph construction. The best performance for static and dynamic graph construction and the highest homophily is **bold**.

| | Initial edges | Model | $k$ | Test MAE ↓ | Test homophily ↑ |
|---|---|---|---|---|---|
| **BL** | - | Mean prediction | - | $6.4090 \pm 0.00$ | - |
| | - | Random Forest | - | $4.1424 \pm 0.01$ | - |
| | - | Linear Regression | - | $3.7545 \pm 0.00$ | - |
| **Static GC** | Random | GCN | - | $6.2158 \pm 0.07$ | $0.742 \pm 0.10$ |
| | | SAGE | - | $\mathbf{3.8764 \pm 0.08}$ | $0.742 \pm 0.10$ |
| | | GraphConv | - | $4.2029 \pm 0.16$ | $0.742 \pm 0.10$ |
| | | GAT | - | $6.4034 \pm 0.07$ | $0.742 \pm 0.10$ |
| | $k$-NN Euclidean | GCN | 2 | $4.3351 \pm 0.07$ | $\mathbf{0.916 \pm 0.07}$ |
| | | SAGE | 10 | $4.1780 \pm 0.17$ | $0.844 \pm 0.06$ |
| | | GraphConv | 2 | $4.1979 \pm 0.04$ | $\mathbf{0.916 \pm 0.07}$ |
| | | GAT | 20 | $4.2888 \pm 0.01$ | $0.834 \pm 0.06$ |
| **Dynamic GC** | No edges | GCN | 2 | $4.0257 \pm 0.06$ | $\mathbf{0.865 \pm 0.10}$ |
| | | SAGE | 5 | $3.8882 \pm 0.03$ | $0.754 \pm 0.10$ |
| | | GraphConv | 5 | $3.9741 \pm 0.05$ | $0.840 \pm 0.08$ |
| | | GAT | 2 | $4.1071 \pm 0.07$ | $0.843 \pm 0.11$ |
| | Random | GCN | 2 | $5.1712 \pm 0.20$ | $0.834 \pm 0.13$ |
| | | SAGE | 10 | $\mathbf{3.8811 \pm 0.04}$ | $0.780 \pm 0.09$ |
| | | GraphConv | 10 | $4.1248 \pm 0.30$ | $0.768 \pm 0.09$ |
| | | GAT | 2 | $5.7138 \pm 0.10$ | $0.831 \pm 0.14$ |
| | $k$-NN Euclidean | GCN | 2 | $4.1109 \pm 0.07$ | $0.849 \pm 0.11$ |
| | | SAGE | 20 | $3.9226 \pm 0.13$ | $0.842 \pm 0.07$ |
| | | GraphConv | 2 | $3.9560 \pm 0.09$ | $0.831 \pm 0.11$ |
| | | GAT | 2 | $4.1603 \pm 0.04$ | $0.837 \pm 0.11$ |

## 5.2 Comparison to Other Published Results

With these results, the question arises as to why population graphs have been believed to improve the performance of medical downstream tasks. We compare our results to published results in the most closely related works, investigating the different performances of baseline models and GNNs on different datasets. We

compare all datasets that have been used in related works: TADPOLE, ABIDE, and UKBB brain age datasets. The related works we pick for comparison are works introducing the concept of population graphs (Parisot et al., 2017), as well as new graph learning techniques that have been applied to or designed for population graph studies (Kazi et al., 2019; 2022; Bintsi et al., 2023a). The results are summarised in Table 5. All our baselines outperform the published baselines in the related works, while our GNN implementations match the performances reported in the respective works. This corroborates our hypothesis that our implementation is on par with previously reported works, while these works seem to underestimate the baseline performance.

Table 5: Comparison of our results to results from related works: Parisot et al. (2017) [1], Kazi et al. (2022) [2], Kazi et al. (2019) [3], and Bintsi et al. (2023a) [4]. The overall best result for each dataset is underlined. The baseline for the UKBB brain age dataset is a ridge regression for our work and a linear regression for the results from Bintsi et al. (2023a); for the TADPOLE dataset: Linear classifier for results from Kazi et al. (2022), random forest for our results; for ABIDE: Ridge regression for results from Parisot et al. (2017), random forest for our results. All our baselines outperform reported baselines in other works, while our GNN implementations match performance.

| Dataset | Score | Method | Convolution | Other reported results | Our results |
|---|---|---|---|---|---|
| **TADPOLE** | Accuracy ↑ | Baseline | - | $0.7022 \pm 0.06$ [2] | $\mathbf{0.9474 \pm 0.00}$ |
| | | dDGM [2] | GCN | $\mathbf{0.9414 \pm 0.02}$ [2] | $0.9333 \pm 0.01$ |
| | | InceptionGCN [3] | InceptionGCN | $0.8435 \pm 0.07$ [3] | |
| **UKBB Brain Age** | MAE ↓ | Baseline | - | $3.82$ [4] | $\mathbf{3.5063 \pm 0.00}$ |
| | | dDGM [2] | GCN | $\mathbf{3.72}$ [4] | $3.8287 \pm 0.03$ |
| | | dDGM [2] | SAGE | - | $\mathbf{3.5034 \pm 0.06}$ |
| | | adaptive [4] | GCN | $3.62$ [4] | - |
| **ABIDE** | Accuracy ↑ | Baseline | - | $0.668$ [1] | $\mathbf{0.7040 \pm 0.01}$ |
| | | Similarity Score [1] | GCN | $0.695$ [1] | - |
| | | InceptionGCN [3] | InceptionGCN | $0.6923 \pm 0.07$ [3] | - |

The discrepancy in baseline performance can partially be due to different models, different hyperparameters, or the utilisation of only a subset of the features for the evaluation of the baselines. Some works, for example, only use the node features of the GNN as input for the baseline, while using additional features for the edge construction of the population graph. We deem this to be an unfair comparison and always use all features that we use for graph construction and as node features as input for the baseline. For the evaluation of the baseline methods on the benchmark citation network datasets, we use only the node features of the graphs since the edges cannot be incorporated in the same feature vector in a straightforward way. Some works do not specify on which features the baseline is evaluated (Parisot et al., 2017).

### 5.3 Population Graphs for Multi-Modal Data Integration

One highly emphasised advantage of population graphs is their utilisation for multi-modal data integration (Parisot et al., 2017; Zheng et al., 2022; Keicher et al., 2021). In one of the first utilisations of population graphs (Parisot et al., 2017), for instance, a graph construction method is introduced that uses clinical features to generate the edges between subjects, while image-derived features are used as node features in the graph. In later approaches, especially for dynamic graph construction, methods moved away from a clear separation between clinical and image-derived features (Kazi et al., 2022). In this so far typical setting of population graphs, we scrutinise this claimed advantage and argue that all available features can easily be appended and, therefore, incorporated into the node features. However, we see exceptions when the information used for edge construction cannot be used as node features. This is the case when high dimensional data is used as node features –e.g. text, audio data or images. However, this setup comes with large memory requirements and has not been studied in detail. We encourage a more critical assessment of the utilisation of GNNs for multi-modal data integration in conventional configurations of population graphs and advocate a shift towards more advanced settings and a more suitable usage of multi-modal data integration for cases where it is indeed beneficial.

We perform several experiments investigating whether GNNs are useful for multi-modal data integration for population graphs. We take the two UKBB datasets and evaluate the performance of GNNs with different combinations of imaging and non-imaging features for graph construction and as node features. The results

are summarised in Table 6. Given that the convolutions GraphSAGE and GraphConv performed best in our previous experiments on population graphs, we limit these results to those two convolutions. The best performing GNN is highlighted in **bold**, the second best in purple, and the third best in green. The corresponding homophily values for each graph structure for both datasets are summarised in Table 7. For these experiments with static graph construction, we experiment with a different model architecture consisting of only one graph convolutional layer, followed by an MLP.

We observe that for the brain age dataset, the best GNN is the one that uses all available features as node features and for edge construction. The second and third-best GNNs also use all features as node features. For the cardiac dataset, the best and second-best models also use all features as node features. However, the third-best model uses only the imaging features as node features and the non-imaging features for edge construction. Furthermore, on the UKBB brain age dataset, some GNNs outperform the respective baseline (which only uses the node features) by small margins. This is not the case for the cardiac dataset. Here none of the GNNs outperform the respective baselines. Interestingly, on the UKBB brain age dataset, the static graph construction results in better performance than dynamic graph construction, which is the opposite for the cardiac dataset. We can also see that the node features slightly dominate the prediction, such that the performance of the GNN somewhat matches the performance of the baseline that uses the node features only. This is reasonable since the specific features used for edge construction are reduced into a simple "measure of similarity". However, overall, the baselines perform on par with the GNNs.

The graph metrics for the experiments are summarised in Tables 3 and 4. The homophily values of the graph structures constructed from different combinations of image and non-image features for the UKBB brain age and cardiac dataset are summarised in Table 7. We can see that for both datasets, all graph structures have similar homophily values, which might be why the performance of all graph structures is very similar when using all node features.

Table 6: Results of different combinations of image-derived and non-imaging features as node features and for graph construction on the UKBB brain age and cardiac datasets. For the age prediction dataset, the baseline is a ridge regression, and for the cardiac dataset, a random forest. GNN outperforms their corresponding node-feature-baseline are underlined. Best GNN: **bold**, second best GNN: purple, third best GNN: green. All scores are evaluated on the test set.

| | Features | | Model | UKBB Brain Age Test MAE ↓ | | UKBB Cardiac Test accuracy ↑ | |
|---|---|---|---|---|---|---|---|
| **Baseline** | - | | Naive baseline | 6.4090 | | 0.5000 | |
| | Non-imaging | | Best baseline | 4.6509 ± 0.00 | | 0.6678 ± 0.00 | |
| | Imaging | | | 3.5063 ± 0.00 | | 0.6969 ± 0.01 | |
| | All | | | 3.4185 ± 0.00 | | **0.7046 ± 0.01** | |
| | Node Feat. | (Initial) Edges | Model | dDGM MAE ↓ | Static MAE ↓ | dDGM acc. ↑ | Static acc. ↑ |
| **Graph Neural Networks** | All | All | GraphSAGE | 3.5034 ± 0.06 | 3.4351 ± 0.00 | 0.6816 ± 0.01 | 0.6609 ± 0.02 |
| | | | GraphConv | 3.5407 ± 0.04 | **3.3524 ± 0.06** | 0.6785 ± 0.01 | 0.6705 ± 0.01 |
| | All | Imaging | GraphSAGE | 3.5471 ± 0.02 | 3.4249 ± 0.00 | **0.6839 ± 0.01** | 0.6739 ± 0.01 |
| | | | GraphConv | 3.5221 ± 0.03 | 3.3758 ± 0.05 | 0.6690 ± 0.01 | 0.6743 ± 0.01 |
| | All | Non-imaging | GraphSAGE | 3.5317 ± 0.04 | 3.4175 ± 0.00 | 0.6724 ± 0.01 | 0.6632 ± 0.01 |
| | | | GraphConv | 3.6792 ± 0.25 | 3.4330 ± 0.01 | 0.6751 ± 0.01 | 0.6644 ± 0.02 |
| | Imaging | Imaging | GraphSAGE | 3.9226 ± 0.13 | 3.7716 ± 0.04 | 0.6743 ± 0.01 | 0.6705 ± 0.00 |
| | | | GraphConv | 3.9560 ± 0.09 | 3.8368 ± 0.00 | 0.6632 ± 0.01 | 0.6628 ± 0.01 |
| | Imaging | Non-imaging | GraphSAGE | 3.9130 ± 0.05 | 3.6791 ± 0.01 | 0.6567 ± 0.01 | 0.6785 ± 0.00 |
| | | | GraphConv | 3.9835 ± 0.01 | 3.7099 ± 0.04 | 0.6805 ± 0.01 | 0.6483 ± 0.01 |
| | Non-imaging | Imaging | GraphSAGE | 4.6767 ± 0.06 | 4.9382 ± 0.00 | 0.6755 ± 0.01 | 0.6521 ± 0.01 |
| | | | GraphConv | 4.0376 ± 0.12 | 5.0410 ± 0.02 | 0.6579 ± 0.01 | 0.6452 ± 0.01 |

## 5.4 Further Components of Impact on Model Performance

In this section, we investigate the impact of the graph structure on model performance from three further viewpoints. (1) The experiments above have indicated that the graph structure has a different impact on different graph convolutions, (2) the complexity of the dataset plays an important role in the performance of GNNs on low-homophily graphs, and (3) if a meaningful graph structure is available, GNNs out-perform

Table 7: Homophily values of the **UKBB brain age** and **cardiac** datasets with $k = 5$ and $k$-NN graph construction, when using all features, only imaging, or only non-imaging features for graph construction.

| Dataset | Features | Homophily |
|---|---|---|
| **Brain Age** | All | $0.8571 \pm 0.07$ |
| | Imaging | $\mathbf{0.8619 \pm 0.07}$ |
| | Non-imaging | $0.8237 \pm 0.08$ |
| **Cardiac** | All | $0.6404 \pm 0.22$ |
| | Imaging | $0.6396 \pm 0.22$ |
| | Non-imaging | $\mathbf{0.6649 \pm 0.23}$ |

graph-agnostic models. Therefore, we perform additional experiments on synthetically generated graph structures at different homophily values. Here, the graph is constructed statically and to specifically match a certain homophily value by using the labels and connecting each node to a specific number of same and differently labelled neighbours. The results for three datasets are visualised in Figure 4, and more visualisations can be found in the appendix in Figure 6.

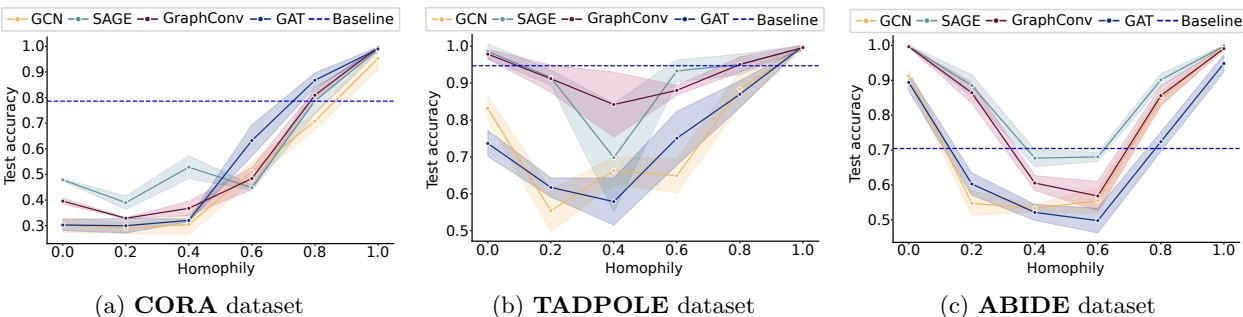

(a) **CORA** dataset      (b) **TADPOLE** dataset      (c) **ABIDE** dataset

Figure 4: Performance of different graph convolutions on synthetic graph structures with different homophily values on (a) the **CORA** dataset, (b) the **TADPOLE** dataset, and (c) the ABIDE dataset. The dashed blue line indicates the mean performance of the best baseline for each dataset.

**Different Types of Graph Convolution** Zhu et al. (2020) have shown interesting correlations between homophily and different graph convolutions. They showed that the separate handling of node features of the node of interest ($x_i$) and its neighbouring nodes ($\mathcal{N}_i$) improves the performance of GNNs on heterophilic graphs. The same accounts for networks that evaluate the $k$-hop neighbourhoods separately. Graph convolutional networks (GCNs) (Kipf & Welling, 2016) do not separate node features of $i$ and $\mathcal{N}_i$, but average the message passing over both in one step (Equation 5). GraphSAGE and GraphConv, on the other hand, distinguish between $x_i$ and $x_j, j \in \mathcal{N}_i$ (Equations 6 and 7). GAT (Equation 8) learns different attention coefficients for $x_i$ and $x_j, j \in \mathcal{N}_i$. However, the network weights are shared for both, which might negatively impact performance on graphs with low homophily. Our experiments support these findings. We observe that the graph structure strongly affects GCN and GAT, whereas GraphSAGE or GraphConv networks perform more consistently across different graph structures.

**Impact of Dataset Complexity** The impact of the homophily on the model performance is not only dependent on the graph convolution but also varies depending on the dataset, probably related to the number of classes in the dataset as well as class imbalance. In order to investigate this, we perform experiments with synthetic graph structures on the TADPOLE dataset (3 classes), the CORA dataset (7 classes), the UKBB cardiac dataset (2 classes), and the ABIDE dataset (2 classes) at different synthetically generated homophily values. Figure 4 shows the performance of different graph convolutions on 3-layer GNNs using synthetically generated graphs for the different datasets. For the CORA dataset (Figure 4a), all models perform worse than the baseline with homophily values lower than 0.8. While all graph convolutions are impacted similarly and perform worse than the baseline for low-homophily graphs, SAGE and GraphConv perform better than GAT and GCN. The low-homophily graphs do not allow the model to learn meaningful node feature embeddings since, during the course of training, node features of differently labelled nodes are

averaged and shared, interfering with the model's goal to distinguish different classes. Interestingly, the performance for the TADPOLE dataset (Figure 4b) looks different. We observe similar differences between the graph convolutions. However, we also observe that only at very high and very low homophily values can the GNN outperform the baseline. Everything in between either matches the performance of the baseline or reaches a worse performance. When we now compare the homophily values of the generated graph structures in our experiments on the TADPOLE dataset above, we can see that most of them have a homophily of around 0.7 or 0.8. The ABIDE dataset requires a graph structure with lower homophily to outperform the baseline. However, the same pattern holds that all population graphs constructed in our experiments reached homophily values in the range where the GNNs under-perform or perform on par with the baselines. This potentially explains why the population graphs do not outperform the graph-agnostic baseline models.

Furthermore, the high performance of the GNNs at low homophily values for the population graphs is highly different from that on the CORA dataset. We attribute this to the capability of the GNNs to learn the opposite labels from the majority of the neighbour labels, which we deem impossible for datasets with more classes. To investigate this further, we evaluate the attention of GAT models trained on graphs with different homophily values on the TADPOLE dataset and observe that low-homophily graphs (homophily=0.1) attribute high attention from differently labelled nodes and low attention to same-labelled nodes. The opposite is the case for high-homophily graphs. This allows the model to also perform well on low-homophily graphs on datasets with only a few classes. More details about these experiments can be found in appendix, Section C.3.

**Impact of Dataset Size**   We investigate the impact of the graph size on model performance with two additional experiments: **(a)** We partition the largest population graph dataset –UKBB brain age– into smaller subsets (25%, 50%, and 75% of the original dataset) and **(b)** generate a synthetic dataset at different sizes and compare GNN performances to baselines. The results of the partitioned UKBB brain age dataset **(a)** and the synthetic dataset **(b)** are summarised in Table 8. For the GNN, we use the DGM adaptive graph construction with the $k$-NN initial graph structure and GraphSAGE convolutions. For the baseline, we use a linear regression for the UKBB brain age dataset and a 4-layer MLP for the synthetic dataset. We do not observe a tendency for the dataset size to have an impact on the difference in performance between the GNN and the baseline on the partitioned UKBB dataset. The same holds for the synthetic dataset, even for very large graphs with 30 000 nodes.

| Dataset | Nr. Nodes | Score | GNN | Baseline | Performance Difference |
|---|---|---|---|---|---|
| UKBB brain age 25% | 1841 | MAE | 3.6804 ± 0.08 | **3.5577 ± 0.00** | -0.1227 |
| UKBB brain age 50% | 3362 | MAE | 3.7658 ± 0.18 | **3.5363 ± 0.00** | -0.2295 |
| UKBB brain age 75% | 4884 | MAE | 3.7022 ± 0.11 | **3.4651 ± 0.00** | -0.2371 |
| UKBB brain age 100% | 6406 | MAE | 3.5034 ± 0.06 | **3.4185 ± 0.00** | -0.0849 |
| Synthetic | 5 000 | ACC | 0.7980 ± 0.03 | **0.8164 ± 0.00** | 0.0184 |
| Synthetic | 10 000 | ACC | 0.8100 ± 0.04 | **0.8532 ± 0.00** | 0.0432 |
| Synthetic | 20 000 | ACC | 0.8895 ± 0.01 | **0.9072 ± 0.00** | 0.0177 |
| Synthetic | 30 000 | ACC | 0.9179 ± 0.01 | **0.9413 ± 0.00** | 0.0234 |

Table 8: Performance differences between the baseline and GNNs for different subsets of the original dataset UKBB brain age (reported MAE) and a synthetically generated dataset (reported accuracy ACC). The column *Performance Difference* indicates the performance of the baseline minus the performance of the GNN.

**Ground Truth Graph Structures**   Based on these results, we argue that the graph construction methods currently utilised for population graphs are insufficient. Only a meaningful graph structure that adds additional information to the node features leads to better performance of GNNs compared to baseline methods. To support this, we investigate commonly used graph construction methods for population graph studies on the frequently used benchmark citation datasets CORA, CITESEER, and PUBMED (Yang et al., 2016). They provide a "ground truth" graph structure, which we can evaluate in comparison to the graphs resulting from graph construction methods used for population graph studies. This allows us to investigate how the different graph construction methods perform compared to a given "ground-truth" adjacency matrix. The results of the best-performing GNNs and baselines on all three datasets are summarised in Table 9. The experiments on all benchmark citation network datasets have shown that GNNs can improve performance compared to simple baseline methods. However, even for the CITESEER dataset, a ridge classifier outperforms

all GNN methods and neural sheaf diffusion networks. The results for the CORA dataset are also visualised in Figure 5. Only the usage of the original edges outperforms the baseline methods, while all static and dynamic graph construction methods yield poor results. This supports the hypothesis that the graph construction methods for population graphs do not add relevant information to the node features. More detailed results can be found in the appendix.

Table 9: **Summary of results on benchmark datasets** of different baseline methods and the best GNNs and Neural Sheaf Models.

| Method | CORA | CITESEER | PUBMED |
|---|---|---|---|
| **Random forest** | $0.7788 \pm 0.00$ | $0.7480 \pm 0.01$ | $0.7286 \pm 0.01$ |
| **Ridge** | $0.7860 \pm 0.00$ | $\mathbf{0.7720 \pm 0.00}$ | $0.7350 \pm 0.00$ |
| **Linear/Logistic** | $0.5750 \pm 0.00$ | $0.5600 \pm 0.00$ | $0.7310 \pm 0.00$ |
| **GNN $k$-NN** | $0.7692 \pm 0.01$ | $0.6908 \pm 0.01$ | $0.6908 \pm 0.01$ |
| **GNN orig. edges** | $0.8540 \pm 0.01$ | $0.7548 \pm 0.01$ | $0.8760 \pm 0.01$ [1] |
| **Neural Sheaf** | $\mathbf{0.8730 \pm 0.01}$ [3] | $0.7714 \pm 0.02$ [3] | $\mathbf{0.8949 \pm 0.00}$ [3] |

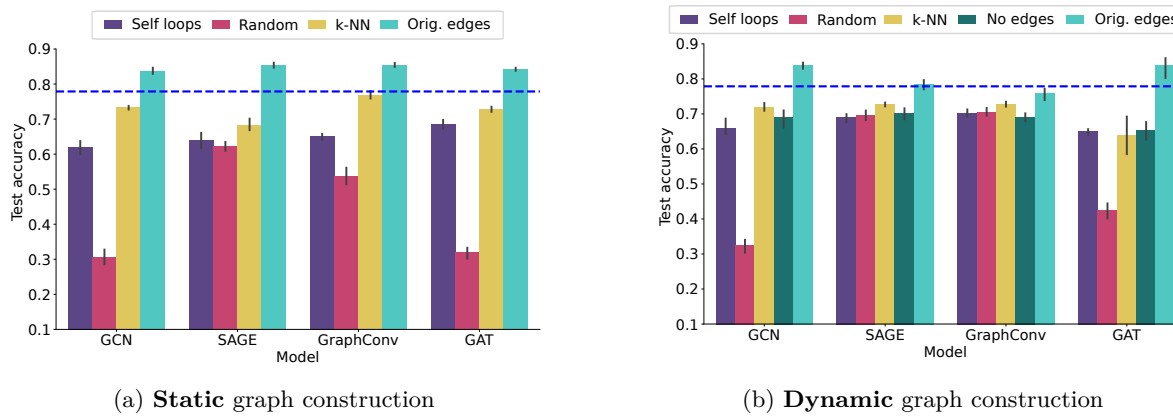

(a) **Static** graph construction         (b) **Dynamic** graph construction

Figure 5: **Results on the CORA dataset** with static (left) and dynamic (right) graph construction.

## 6 Discussion

In this work, we evaluate the performance of medical population graphs on five population graph datasets and compare state-of-the-art graph learning techniques to well-tuned baseline models. We consistently observe the following findings:

1. **GCN and GAT are poorly suited for population graph studies.** GNNs using GraphSAGE and GraphConv convolutions consistently outperform GCN and GAT models, which leads to the conclusion that the latter methods are unsuitable for GNNs in population graph studies. We attribute this to the fact that GCN and GAT networks are highly affected by the graph structure, whereas GraphSAGE and GraphConv networks are more robust in this regard. This also manifests in the fact that GCN and GAT networks benefit more from dynamic graph construction than the other two convolutions and that GraphSAGE and GraphConv models can perform equally well on random graph structures.

2. **The utilisation of population graphs with the goal of multi-modal data integration might not be as promising as believed.** The most frequently used method for the construction of population graphs includes a separation of features into node features and ones utilised for edge construction. We show that using all available features for edge construction and as node features might lead to better results and argue that a concatenation of all features is easily doable –except

when using images as node features. We see potential in using population graphs in different settings where the connectivity information cannot easily be integrated with the node features.

3. **None of the state-of-the-art GNN methods significantly outperform well-tuned baseline methods** (see Table 2). This raises the question of whether population graphs –in the way they are currently used– have any benefit over graph-agnostic models. In Section 5.4, we investigate the interplay of the graph structure and the performance of the GNNs on a population graph dataset and conclude that only a nearly perfect graph structure leads to a better performance of GNNs compared to baseline models, which has not been possible with the current graph construction methods.

4. **Better graph construction methods are required.** The experiments on the benchmark datasets and the synthetically generated graph structures with different homophily values (see Figures 4 and 5) show that GNNs can improve downstream task performance if the graph structure is "meaningful". However, current graph construction methods do not lead to valuable graph structures, which makes graph construction the performance bottleneck in these settings. The same is represented by the fact that random graph structures often achieved comparable results to approaches like $k$-NN graphs.

We furthermore note that all baseline models are easy to implement using standard libraries such as `scikit-learn` (Pedregosa et al., 2011), are significantly faster to fit than the training of GNNs, and do not require extensive hyperparameter tuning.

## 7 Conclusion and Future Work

Medical population graphs were first introduced by Parisot et al. (2017) to allow for a population-wide representation of a cohort of patients. The idea behind the utilisation of population graphs is that subjects that share similar phenotypes (and are therefore neighbours in the population graph), also show similar pathologies. Thus, the neighbouring nodes are hoped to improve model performance when using graph deep learning methods. They have since then been combined with GNNs and used on multiple medical datasets. Most works utilise population graphs as a method for multi-modal data integration (Parisot et al., 2017; Kazi et al., 2019; Cosmo et al., 2020; Bintsi et al., 2023a). Here, a subset of the features are used as node features (usually imaging features), while other features (usually non-imaging) are used to generate the graph structure (the edges).

In this work, we perform an extensive study on how GNNs are used in the context of population graphs and compare different graph-learning methods to graph-agnostic baseline models. We use five medical population graph datasets, including all publicly available datasets used for population graph studies in related works. We utilise state-of-the-art (a) static graph construction methods, (b) dynamic graph construction methods, and (c) neural sheaf diffusion models. The latter have been designed to address two of the most dominant problems of GNNs: over-smoothing and performance on low-homophily graphs. We investigate the usage of neural sheaf diffusion models since the graph construction methods for population graphs seem to result in unideal graph structures, which might benefit from the use of neural sheaf diffusion models.

Even though we reach comparable results to related works on population graphs with GNNs for all methods, none of the GNNs significantly out-perform the strongest baseline method. This raises the question of how powerful population graphs indeed are and whether they are a suitable data representation combined with GNNs. We conclude that currently available graph construction methods are the performance bottleneck of GNNs on population graphs compared to graph-agnostic methods. We see a need for either more advanced methods to learn a graph structure that contains additional meaningful information to the node features or novel ideas on how to build population graphs from additional information that cannot be represented as node features. When using synthetically generated graph structures, we observe that only graphs with higher homophily than possible to extract from the node features result in better performance of GNNs compared to properly tuned graph-agnostic methods such as a random forest or linear regression (Figure 4). Even a dynamic graph construction method, which optimises the graph structure during training, does not reach a "good enough" graph structure. Also, models designed for "low-quality" graph structures (e.g. neural sheaf diffusion models) do not improve performance on population graphs. The fact that our baseline models

outperform the results reported in related works emphasises the importance of appropriate tuning of baseline methods in general. It shows that the currently available graph construction methods for population graphs are insufficient.

There are a few more graph construction methods that we did not evaluate in this work, such as *Similarity Scores*. The first one was introduced by Parisot et al. (2017) and followed by several extensions and modifications (Ghorbani et al., 2022; Vivar et al., 2021; Pellegrini et al., 2022; Peng et al., 2022; Lu et al., 2022). In this work, we focus on using *k*-NN graphs since this method has been shown to achieve the best results in related works (Bintsi et al., 2023b) and preliminary experiments. Furthermore, investigating other graph convolutions or different GNN architectures in combination with specific population graph setups might give more insights. One example would be higher-order GNNs for node-level predictions (Li et al., 2021). We would see this as a fitting method for longitudinal studies. Finally, it could be interesting to evaluate additional graph assessment metrics (Luan et al., 2021; Xie et al., 2020; Luan et al., 2022) and their correlation with graph construction methods and model performance.

We generally see three future directions for population graph studies. Either (a) new and better graph construction methods need to be developed for population graphs to bring benefits to medical downstream tasks, (b) innovative applications of population graphs that truly benefit from the usage of connectivity information need to be explored, or (c) the usage of population graphs in combination with GNNs does not seem valuable for the performance of medical downstream tasks. It would be interesting to follow up with a theoretical analysis. Our experiments showed that the graph construction is a major performance bottleneck for population graphs. We believe this to be a good starting point for follow-up analyses. For better graph construction methods, we see the requirement of increasing the information content of the graph structure compared to the node features alone. This could potentially be achieved by encoding information in the graph structure that cannot be trivially added to the node features, such as genetic similarity between subjects or the risk groups in survival analysis. Other potentially interesting applications of population graphs, where the edges add additional information to the node features, are geospatial graphs. This could include analysing location-based health data, disease spreading, or tracing local differences in medication or care units. Also, time-series data has not been explored in great detail in the context of population graphs, which might add more valuable information to the graph structure. The concept of population graphs has, with a few exceptions (Keicher et al., 2021), mostly focused on vector data instead of images. This is mostly because image data is much larger and, therefore, more difficult to fit into memory in the context of population graphs. This could be another interesting future direction to improve the predictive power of population graphs.

### Acknowledgements

TM and SS were supported by the ERC (Deep4MI - 884622). This work has been conducted under the UK Biobank applications 87802 and 18545. SS has furthermore been supported by BMBF and the NextGenerationEU of the European Union. AZ and GK were supported by the German Ministry of Education and Research (BMBF) under Grant Number 01ZZ2316C (PrivateAIM). The data collection and sharing of the TADPOLE dataset was funded by the Alzheimer's Disease Neuroimaging Initiative (ADNI) (National Institutes of Health Grant U01 AG024904) and DOD ADNI (Department of Defense award number W81XWH-12-2-0012).

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

## A  Additional Information on the Datasets

We here provide some additional information on some of the population graph datasets.

**TADPOLE**  For the TADPOLE dataset, we follow the approach from Kazi et al. (2022) and use the same features as in their work.

**ABIDE**  For the ABIDE dataset, we follow the approach from Parisot et al. (2017) and use the following non-imaging features: Sex and site. The imaging features are extracted in the same way as in their work.

**UKBB cardiac**  We use the following non-imaging features from the UKBB: Age, sex, body fat percentage, smoking status, body mass index, and the frequency of exercises in the last four weeks. The imaging features are extracted from these subjects' cardiac magnetic resonance images (MRIs) and contain information such as end-diastolic, end-systolic volume, stroke volume, and ejection fraction for both ventricles and myocardial-wall thickness. More information about the imaging features can be found in Bai et al. (2020).

**COVID**  The COVID dataset is an in-house dataset, with the task of predicting whether a CoViD patient will require an intensive care unit. The non-imaging features are demographics, blood values, and prior diseases such as age, sex, fever, coughing, the loss of taste or smell, other symptoms, immunosuppressors, duration of symptomatic, shortness of breath, GIT symptoms, neurological symptoms, acute, prior diseases, temperature, oxygen saturation.

**UKBB brain age**  For this dataset, we follow the approach from Bintsi et al. (2023a) for both imaging and non-imaging features. We use the same non-imaging features as in the original work: Sex, weight, height, body mass index, systolic blood pressure, diastolic blood pressure, college education, smoking status, alcohol intake frequency, stroke, diabetes, walking per week, moderate exercising per week, vigorous exercising per week, fluid intelligence, tower rearranging: number of puzzles correct, trail making task: duration to complete numeric path trail 1, trail making task: duration to complete alphanumeric path trail 2, matrix pattern completion: number of puzzles correctly solved, matrix pattern completion: duration spent answering each puzzle.

**Synthetic dataset**  We generate a synthetic dataset using `sklearn` with 4 classes and a varying number of nodes to investigate the impact of the dataset size on the GNN performance. We use 50 node features, of which 10 are informative.

## B  Hyperparameters and Model Architectures

We summarise the hyperparameter ranges used for the sweeps for our experiments in Table 10. We distinguish between experiments using static graph construction, dynamic graph construction, and baseline tuning.

## C  Additional Results

We here summarise the results of additional experiments to the ones reported in the main text on the datasets CORA (Table 11), TADPOLE (Table 12), UKBB brain age (Table 13), and UKBB cardiac (Table 14). For example, the performance of the GNNs on an imitated graph structure that only contains self-loops simulates transductive learning without a meaningful graph structure and a graph construction using $k$-NN with the cosine distance. All here summarised experiments follow the same setup as introduced in Section 4.4. Figure 6 visualises more results following the same approach as in Section 5.4 with the additional dataset UKBB cardiac and larger.

### C.1  Benchmark Datasets

With the experiments on the CORA dataset (Table 11), we observe that only the GNNs that utilise the "ground truth" edges out-perform our baseline methods, while the commonly used graph construction methods

|  | Parameter | Range |
|---|---|---|
| **All** | Learning rate | [0.00001; 0.09] |
|  | Dropout | [0.0,0.1,0.2,0.3,0.4] |
|  | $k$ | [2,5,10,20] |
|  | Convolutions | [GAT, GCN, GraphConv, GraphSAGE] |
| **St.** | Nr. layers | [1,2,3] |
|  | Hidden channels | 32 |
| **Dyn.** | FC layers | [[32,8,1], [8,8,3]] |
|  | DGM layers | [[[32,16,4]], [[32,16,4],[],[]]] |
|  | Conv layers | [[[32,32]], [[32,32],[32,16],[16,8]]] |
| **Neural Sheaf** | d | [2,3,4] |
|  | Add lp | [0,1] |
|  | Add hp | [0,1] |
|  | Nr. layers | [2,3,4,5,6] |
|  | Hidden channels | [8,16,32] |
|  | Input dropout | [0.0, 0.1, 0.2, 0.3, 0.4, 0.5, 0.6, 0.7, 0.8, 0.9] |
|  | Dropout | [0.0, 0.1, 0.2, 0.3, 0.4, 0.5, 0.6, 0.7, 0.8, 0.9] |
|  | Learning rate | [0.02, 0.01, 0.05, 0.05, 0.001] |
|  | Sheaf type | [BundleSheaf, DiagSheaf, GeneralScheaf] |
| **RF** | Max depth | [2;20] |
|  | Nr. estimators | [500;2000] |
| **R** | Alpha | [0.001; 50] |

Table 10: Hyperparameter ranges for static and dynamic graph construction experiments. **All GNN**: for all GNN experiments, **St.**: parameters for static experiments only, **Dyn.**: for dynamic experiments only, **Neural Sheaf**: experiments with neural sheaf diffusion models, **RF**: random forest experiments, **R**: ridge classifier/regressor experiments.

for population graphs also do not benefit performance on the CORA dataset. We observed similar results for the other benchmark datasets.

## C.2 Population Graph Datasets

Tables 12, 13, and 14 show the results of additional experiments on the TADPOLE, UKBB brain age, and UKBB cardiac datasets, respectively. We here also test the performance of GNNs on a graph that only contains self-loops, which mimics a transductive learning setting without actually using a graph structure. This rules out that a potential performance increase of GNNs stems from the fact that all node features are seen during training, which is not the case for standard baseline models, such as random forests or linear regressions. However, we here also do not observe an improved performance of GNNs compared to our baseline models.

## C.3 Attention Evaluation

In Section 5.4, we observed an impact of the dataset complexity on GNN performance at different homophily values. While for the CORA dataset, which has 7 classes, low-homophily graphs always resulted in poor performance, on the TADPOLE dataset, low-homophily graphs were also able to lead to good GNN performance. Similar to Figure 4, we visualise an additional dataset in Figure 6. We attribute the relatively good performance of all models at low homophily values on the TADPOLE dataset (Figure 6b) to the learning of opposite labels for specific node features. If most of the neighbouring nodes share a different label than the one the node of interest holds, but this is consistent across the graph –the graph has a low CCNS distance–, then the network can still learn to make the correct predictions. We show this by evaluating the attention values of GAT networks of four synthetic graph structures with different homophily values. All values are summarised in the appendix in Table 15. We always report the normalised sum of all attention heads of the GAT. At homophily 0.9 (where most neighbours share the same label as the node of interest), the attention from the neighbours with the same label is the highest. On the other hand, at hom = 0.5, all nodes receive the highest attention from neighbours with class label "MCI". This makes it very difficult for the network to distinguish between nodes of different labels, and therefore to make the correct predictions. At very low homophily (hom = 0.1), the attention of the neighbours with the same label is 0, which again,

Table 11: Results of the experiments on the **CORA** dataset. BL: baselines, $k$: number of neighbours, Transd.: transductive learning with only self-loops. GNNs out-performing the BL are underlined, and the best performances of static and dynamic graph constructions, the highest homophily and the lowest CCNS distance are **bold**.

| | Initial edges | Model | $k$ | Test acc | Homophily ↑ Train | Homophily ↑ Test | CCNS distance ↓ Train | CCNS distance ↓ Test |
|---|---|---|---|---|---|---|---|---|
| **BL** | - | Random Forest | - | 0.7788 ± 0.00 | - | - | - | - |
| | | Ridge classifier | - | 0.7860 ± 0.00 | - | - | - | - |
| | | MLP | - | 0.6030 ± 0.00 | - | - | - | - |
| **Transd.** | Self-loops | GCN | - | 0.6200 ± 0.02 | 1.000 ± 0.00 | 1.000 ± 0.00 | 0.000 | 0.000 |
| | | SAGE | - | 0.6396 ± 0.03 | 1.000 ± 0.00 | 1.000 ± 0.00 | 0.000 | 0.000 |
| | | GraphConv | - | 0.6504 ± 0.01 | 1.000 ± 0.00 | 1.000 ± 0.00 | 0.000 | 0.000 |
| | | GAT | - | 0.6848 ± 0.01 | 1.000 ± 0.00 | 1.000 ± 0.00 | 0.000 | 0.000 |
| **Static graph construction** | Random | GCN | - | 0.3068 ± 0.02 | 0.171 ± 0.26 | 0.201 ± 0.29 | 0.373 | 0.356 |
| | | SAGE | - | 0.6224 ± 0.02 | 0.171 ± 0.26 | 0.201 ± 0.29 | 0.373 | 0.356 |
| | | GraphConv | - | 0.5388 ± 0.03 | 0.171 ± 0.26 | 0.201 ± 0.29 | 0.373 | 0.356 |
| | | GAT | - | 0.3208 ± 0.02 | 0.171 ± 0.26 | 0.201 ± 0.29 | 0.373 | 0.356 |
| | $k$-NN Euclidean | GCN | 20 | 0.7336 ± 0.01 | 0.498 ± 0.23 | 0.495 ± 0.22 | 0.378 | 0.396 |
| | | SAGE | 20 | 0.6836 ± 0.02 | 0.498 ± 0.23 | 0.495 ± 0.22 | 0.378 | 0.396 |
| | | GraphConv | 20 | 0.7692 ± 0.01 | 0.498 ± 0.23 | 0.495 ± 0.22 | 0.378 | 0.396 |
| | | GAT | 20 | 0.7288 ± 0.01 | 0.498 ± 0.23 | 0.495 ± 0.22 | 0.378 | 0.396 |
| | $k$-NN Cosine | GCN | 20 | 0.7332 ± 0.01 | 0.537 ± 0.24 | 0.537 ± 0.23 | 0.344 | 0.362 |
| | | SAGE | 20 | 0.6668 ± 0.01 | 0.537 ± 0.24 | 0.537 ± 0.23 | 0.344 | 0.362 |
| | | GraphConv | 20 | 0.7628 ± 0.01 | 0.537 ± 0.24 | 0.537 ± 0.23 | 0.344 | 0.362 |
| | | GAT | 20 | 0.7260 ± 0.01 | 0.537 ± 0.24 | 0.537 ± 0.23 | 0.344 | 0.362 |
| | Orig. edges | GCN | - | 0.8332 ± 0.01 | **0.830 ± 0.29** | **0.860 ± 0.29** | **0.101** | **0.084** |
| | | SAGE | - | **0.8540 ± 0.01** | **0.830 ± 0.29** | **0.860 ± 0.29** | **0.101** | **0.084** |
| | | GraphConv | - | **0.8540 ± 0.01** | **0.830 ± 0.29** | **0.860 ± 0.29** | **0.101** | **0.084** |
| | | GAT | - | 0.8420 ± 0.00 | **0.830 ± 0.29** | **0.860 ± 0.29** | **0.101** | **0.084** |
| **Dynamic graph construction** | No edges | GCN | 2 | 0.6900 ± 0.03 | **0.987 ± 0.10** | 0.749 ± 0.42 | 0.072 | 0.181 |
| | | SAGE | 2 | 0.7000 ± 0.02 | 0.589 ± 0.38 | 0.510 ± 0.37 | 0.232 | 0.267 |
| | | GraphConv | 2 | 0.6904 ± 0.01 | 0.880 ± 0.21 | 0.769 ± 0.25 | 0.085 | 0.144 |
| | | GAT | 2 | 0.6532 ± 0.03 | 0.921 ± 0.20 | 0.652 ± 0.43 | **0.050** | 0.208 |
| | Self-loops | GCN | 5 | 0.5932 ± 0.13 | 0.737 ± 0.31 | 0.612 ± 0.37 | 0.176 | 0.244 |
| | | SAGE | 20 | 0.6900 ± 0.01 | 0.857 ± 0.23 | 0.751 ± 0.25 | 0.092 | 0.160 |
| | | GraphConv | 2 | 0.7024 ± 0.01 | 0.696 ± 0.27 | 0.586 ± 0.32 | 0.185 | 0.273 |
| | | GAT | 5 | 0.6492 ± 0.01 | 0.796 ± 0.28 | 0.584 ± 0.39 | 0.139 | 0.259 |
| | Random | GCN | 2 | 0.3240 ± 0.02 | 0.663 ± 0.28 | 0.230 ± 0.38 | 0.201 | 0.351 |
| | | SAGE | 10 | 0.6960 ± 0.01 | 0.674 ± 0.25 | 0.534 ± 0.32 | 0.206 | 0.323 |
| | | GraphConv | 2 | 0.7052 ± 0.01 | 0.831 ± 0.24 | 0.719 ± 0.25 | 0.101 | 0.180 |
| | | GAT | 10 | 0.4252 ± 0.02 | 0.405 ± 0.23 | 0.252 ± 0.23 | 0.436 | 0.544 |
| | $k$-NN Euclidean | GCN | 5 | 0.7192 ± 0.01 | 0.581 ± 0.31 | 0.533 ± 0.30 | 0.314 | 0.363 |
| | | SAGE | 5 | 0.7264 ± 0.01 | 0.838 ± 0.23 | 0.676 ± 0.35 | 0.097 | 0.222 |
| | | GraphConv | 5 | 0.7284 ± 0.01 | 0.884 ± 0.21 | 0.801 ± 0.24 | 0.073 | **0.129** |
| | | GAT | 20 | 0.6388 ± 0.06 | 0.419 ± 0.27 | 0.415 ± 0.28 | 0.429 | 0.446 |
| | $k$-NN Cosine | GCN | 5 | 0.7424 ± 0.00 | 0.611 ± 0.33 | 0.570 ± 0.32 | 0.299 | 0.349 |
| | | SAGE | 5 | 0.7216 ± 0.01 | 0.774 ± 0.26 | 0.663 ± 0.35 | 0.153 | 0.234 |
| | | GraphConv | 5 | 0.7304 ± 0.01 | 0.890 ± 0.21 | 0.778 ± 0.25 | 0.070 | 0.143 |
| | | GAT | 20 | 0.6716 ± 0.01 | 0.662 ± 0.30 | 0.634 ± 0.37 | 0.216 | 0.236 |
| | Orig. edges | GCN | 20 | **0.8372 ± 0.01** | 0.861 ± 0.24 | **0.813 ± 0.31** | 0.086 | 0.133 |
| | | SAGE | 10 | 0.7832 ± 0.01 | 0.958 ± 0.10 | 0.780 ± 0.32 | 0.019 | 0.138 |
| | | GraphConv | 2 | 0.7576 ± 0.02 | 0.819 ± 0.25 | 0.780 ± 0.29 | 0.115 | 0.149 |
| | | GAT | 2 | 0.8388 ± 0.04 | 0.885 ± 0.21 | 0.807 ± 0.29 | 0.071 | 0.131 |

makes it possible for the network to distinguish nodes by their neighbourhood, enabling correct predictions. Three examples of 2-hop neighbourhoods at the different homophily values are visualised in Figure 7. The label is indicated by the node colour and the distance between two nodes indicates the attention value of this edge. While at hom = 0.9, most neighbours share the same label, at a low homophily value of 0.1 (c), most neighbours have a different label and the attention values are similar across them. At an in-between homophily of 0.4, several nodes share the same label, while others do not.

Table 12: Results of the experiments on the **TADPOLE** dataset. BL: baselines, $k$: number of neighbours, Transd.: transductive learning with only self-loops. Overall, the best performance for static and dynamic graph construction is underlined, the best performance for static and dynamic graph construction, highest homophily and lowest DNNS distance are **bold**.

| | Initial edges | Model | $k$ | Test acc ↑ | Homophily ↑ Train | Test | CCNS distance ↓ Train | Test |
|---|---|---|---|---|---|---|---|---|
| **BL** | - | Majority vote | - | $0.5674 \pm 0.00$ | - | - | - | - |
| | - | Random forest | - | $\mathbf{0.9474 \pm 0.00}$ | - | - | - | - |
| | - | Logistic regression | - | $0.8597 \pm 0.00$ | - | - | - | - |
| **Transd.** | Self-loops | GCN | - | $0.9018 \pm 0.01$ | $1.000 \pm 0.00$ | $1.000 \pm 0.00$ | 0.000 | 0.000 |
| | | SAGE | - | $0.8772 \pm 0.01$ | $1.000 \pm 0.00$ | $1.000 \pm 0.00$ | 0.000 | 0.000 |
| | | GraphConv | - | $0.8912 \pm 0.01$ | $1.000 \pm 0.00$ | $1.000 \pm 0.00$ | 0.000 | 0.000 |
| | | GAT | - | $0.6386 \pm 0.07$ | $1.000 \pm 0.00$ | $1.000 \pm 0.00$ | 0.000 | 0.000 |
| **Static graph construction** | Random | GCN | - | $0.7965 \pm 0.04$ | $0.457 \pm 0.49$ | $0.426 \pm 0.49$ | 0.350 | 0.348 |
| | | SAGE | - | $0.8877 \pm 0.01$ | $0.457 \pm 0.49$ | $0.426 \pm 0.49$ | 0.350 | 0.348 |
| | | GraphConv | - | $0.8842 \pm 0.01$ | $0.457 \pm 0.49$ | $0.426 \pm 0.49$ | 0.350 | 0.348 |
| | | GAT | - | $0.7930 \pm 0.04$ | $0.457 \pm 0.49$ | $0.426 \pm 0.49$ | 0.350 | 0.348 |
| | $k$-NN Euclidean | GCN | 5 | $0.7439 \pm 0.03$ | $0.754 \pm 0.23$ | $0.775 \pm 0.24$ | 0.283 | 0.213 |
| | | SAGE | 5 | $0.8982 \pm 0.03$ | $0.754 \pm 0.23$ | $0.775 \pm 0.24$ | 0.283 | 0.213 |
| | | GraphConv | 5 | $0.9088 \pm 0.01$ | $0.754 \pm 0.23$ | $0.775 \pm 0.24$ | 0.283 | 0.213 |
| | | GAT | 2 | $0.7895 \pm 0.04$ | $\mathbf{0.857 \pm 0.23}$ | $\mathbf{0.904 \pm 0.20}$ | **0.184** | 0.094 |
| | $k$-NN Cosine | GCN | 5 | $0.7789 \pm 0.02$ | $0.760 \pm 0.23$ | $0.754 \pm 0.25$ | 0.276 | 0.221 |
| | | SAGE | 5 | $0.8877 \pm 0.02$ | $0.760 \pm 0.23$ | $0.754 \pm 0.25$ | 0.276 | 0.221 |
| | | GraphConv | 5 | $\mathbf{0.9333 \pm 0.01}$ | $0.760 \pm 0.23$ | $0.754 \pm 0.25$ | 0.276 | 0.221 |
| | | GAT | 2 | $0.8105 \pm 0.02$ | $0.855 \pm 0.23$ | $0.895 \pm 0.21$ | 0.192 | 0.105 |
| **Dynamic graph construction** | No edges | GCN | 20 | $0.9263 \pm 0.03$ | $0.899 \pm 0.19$ | $0.919 \pm 0.19$ | 0.143 | **0.073** |
| | | SAGE | 20 | $0.9053 \pm 0.02$ | $0.867 \pm 0.20$ | $0.806 \pm 0.21$ | 0.183 | 0.183 |
| | | GraphConv | 2 | $0.9228 \pm 0.02$ | $0.919 \pm 0.18$ | $0.798 \pm 0.34$ | **0.107** | 0.190 |
| | | GAT | 20 | $0.9018 \pm 0.06$ | $0.739 \pm 0.24$ | $0.908 \pm 0.15$ | 0.280 | 0.101 |
| | Self-loops | GCN | 10 | $0.9298 \pm 0.02$ | $0.891 \pm 0.21$ | $0.902 \pm 0.16$ | 0.150 | 0.085 |
| | | SAGE | 5 | $0.9088 \pm 0.02$ | $0.900 \pm 0.19$ | $0.614 \pm 0.29$ | 0.140 | 0.441 |
| | | GraphConv | 5 | $0.9228 \pm 0.02$ | $\mathbf{0.920 \pm 0.18}$ | $\mathbf{0.937 \pm 0.15}$ | 0.113 | 0.051 |
| | | GAT | 20 | $0.9123 \pm 0.05$ | $0.826 \pm 0.24$ | $0.784 \pm 0.21$ | 0.236 | 0.204 |
| | Random | GCN | 2 | $0.8421 \pm 0.06$ | $0.912 \pm 0.20$ | $0.851 \pm 0.27$ | 0.132 | 0.177 |
| | | SAGE | 10 | $0.9228 \pm 0.02$ | $0.834 \pm 0.23$ | $0.423 \pm 0.22$ | 0.205 | 0.616 |
| | | GraphConv | 5 | $0.8947 \pm 0.03$ | $0.775 \pm 0.24$ | $0.411 \pm 0.25$ | 0.273 | 0.594 |
| | | GAT | 5 | $0.8632 \pm 0.02$ | $0.903 \pm 0.20$ | $0.895 \pm 0.20$ | 0.145 | 0.119 |
| | $k$-NN Euclidean | GCN | 2 | $0.9333 \pm 0.01$ | $0.811 \pm 0.25$ | $0.793 \pm 0.28$ | 0.229 | 0.204 |
| | | SAGE | 20 | $0.9368 \pm 0.01$ | $0.896 \pm 0.19$ | $0.461 \pm 0.63$ | 0.138 | 0.632 |
| | | GraphConv | 10 | $0.8947 \pm 0.02$ | $0.736 \pm 0.23$ | $0.777 \pm 0.29$ | 0.302 | 0.219 |
| | | GAT | 10 | $0.9123 \pm 0.03$ | $0.826 \pm 0.24$ | $0.775 \pm 0.29$ | 0.223 | 0.206 |
| | $k$-NN Cosine | GCN | 2 | $0.8421 \pm 0.02$ | $0.833 \pm 0.24$ | $0.786 \pm 0.30$ | 0.210 | 0.199 |
| | | SAGE | 20 | $\mathbf{0.9404 \pm 0.02}$ | $0.822 \pm 0.23$ | $0.899 \pm 0.21$ | 0.220 | 0.084 |
| | | GraphConv | 10 | $0.8982 \pm 0.02$ | $0.740 \pm 0.25$ | $0.761 \pm 0.28$ | 0.304 | 0.213 |
| | | GAT | 10 | $0.8316 \pm 0.04$ | $0.846 \pm 0.23$ | $0.828 \pm 0.27$ | 0.201 | 0.187 |

Table 13: Results of the experiments on the **UKBB Brain Age** dataset. BL: baselines, $k$: number of neighbours, Transd.: transductive training with only self-loops. The best performance and highest homophily for static and dynamic graph construction are **bold**. For all methods, homophily is evaluated on the train and test set.

| | Initial edges | Model | $k$ | Test MAE ↓ | Homophily ↑ | |
| | | | | | Train | Test |
|---|---|---|---|---|---|---|
| **BL** | - | Mean prediction | - | $6.4090 \pm 0.00$ | - | - |
| | - | Random Forest | - | $4.1424 \pm 0.01$ | - | - |
| | - | Linear Regression | - | $3.7545 \pm 0.00$ | - | - |
| **Transd.** | Self-loops | GCN | - | $4.0236 \pm 0.12$ | $1.000 \pm 0.00$ | $1.000 \pm 0.00$ |
| | | SAGE | - | $4.0339 \pm 0.05$ | $1.000 \pm 0.00$ | $1.000 \pm 0.00$ |
| | | GraphConv | - | $3.9750 \pm 0.06$ | $1.000 \pm 0.00$ | $1.000 \pm 0.00$ |
| | | GAT | - | $3.9477 \pm 0.04$ | $1.000 \pm 0.00$ | $1.000 \pm 0.00$ |
| **Static graph construction** | Random | GCN | - | $6.2158 \pm 0.07$ | $0.750 \pm 0.10$ | $0.742 \pm 0.10$ |
| | | SAGE | - | $\mathbf{3.8764 \pm 0.08}$ | $0.750 \pm 0.10$ | $0.742 \pm 0.10$ |
| | | GraphConv | - | $4.2029 \pm 0.16$ | $0.750 \pm 0.10$ | $0.742 \pm 0.10$ |
| | | GAT | - | $6.4034 \pm 0.07$ | $0.750 \pm 0.10$ | $0.742 \pm 0.10$ |
| | $k$-NN Euclidean | GCN | 2 | $4.3351 \pm 0.07$ | $\mathbf{0.915 \pm 0.07}$ | $\mathbf{0.916 \pm 0.07}$ |
| | | SAGE | 10 | $4.1780 \pm 0.17$ | $0.843 \pm 0.06$ | $0.844 \pm 0.06$ |
| | | GraphConv | 2 | $4.1979 \pm 0.04$ | $\mathbf{0.915 \pm 0.07}$ | $\mathbf{0.916 \pm 0.07}$ |
| | | GAT | 20 | $4.2888 \pm 0.01$ | $0.832 \pm 0.06$ | $0.834 \pm 0.06$ |
| | $k$-NN Cosine | GCN | 2 | $4.3808 \pm 0.08$ | $\mathbf{0.915 \pm 0.07}$ | $\mathbf{0.919 \pm 0.06}$ |
| | | SAGE | 10 | $4.2302 \pm 0.21$ | $0.843 \pm 0.06$ | $0.844 \pm 0.06$ |
| | | GraphConv | 2 | $4.2260 \pm 0.06$ | $\mathbf{0.915 \pm 0.07}$ | $\mathbf{0.919 \pm 0.06}$ |
| | | GAT | 20 | $4.3182 \pm 0.03$ | $0.833 \pm 0.06$ | $0.833 \pm 0.06$ |
| **Dynamic graph construction** | No edges | GCN | 2 | $4.0257 \pm 0.06$ | $\mathbf{0.886 \pm 0.09}$ | $\mathbf{0.865 \pm 0.10}$ |
| | | SAGE | 5 | $3.8882 \pm 0.03$ | $0.752 \pm 0.10$ | $0.754 \pm 0.10$ |
| | | GraphConv | 5 | $3.9741 \pm 0.05$ | $0.845 \pm 0.08$ | $0.840 \pm 0.08$ |
| | | GAT | 2 | $4.1071 \pm 0.07$ | $0.840 \pm 0.10$ | $0.843 \pm 0.11$ |
| | Self-loops | GCN | 2 | $3.9869 \pm 0.06$ | $0.844 \pm 0.10$ | $0.841 \pm 0.10$ |
| | | SAGE | 20 | $3.9496 \pm 0.16$ | $0.781 \pm 0.07$ | $0.780 \pm 0.08$ |
| | | GraphConv | 20 | $3.9422 \pm 0.13$ | $0.849 \pm 0.06$ | $0.845 \pm 0.07$ |
| | | GAT | 2 | $4.0825 \pm 0.07$ | $0.844 \pm 0.10$ | $0.839 \pm 0.10$ |
| | Random | GCN | 2 | $5.1712 \pm 0.20$ | $0.837 \pm 0.12$ | $0.834 \pm 0.13$ |
| | | SAGE | 10 | $\mathbf{3.8811 \pm 0.04}$ | $0.769 \pm 0.08$ | $0.780 \pm 0.09$ |
| | | GraphConv | 10 | $4.1248 \pm 0.30$ | $0.770 \pm 0.08$ | $0.768 \pm 0.09$ |
| | | GAT | 2 | $5.7138 \pm 0.10$ | $0.852 \pm 0.11$ | $0.831 \pm 0.14$ |
| | $k$-NN Euclidean | GCN | 2 | $4.1109 \pm 0.07$ | $0.835 \pm 0.10$ | $0.849 \pm 0.11$ |
| | | SAGE | 20 | $3.9226 \pm 0.13$ | $0.845 \pm 0.06$ | $0.842 \pm 0.07$ |
| | | GraphConv | 2 | $3.9560 \pm 0.09$ | $0.843 \pm 0.11$ | $0.831 \pm 0.11$ |
| | | GAT | 2 | $4.1603 \pm 0.04$ | $0.835 \pm 0.10$ | $0.837 \pm 0.11$ |
| | $k$-NN Cosine | GCN | 2 | $4.0975 \pm 0.05$ | $0.839 \pm 0.10$ | $0.844 \pm 0.10$ |
| | | SAGE | 20 | $3.9353 \pm 0.12$ | $0.837 \pm 0.06$ | $0.837 \pm 0.07$ |
| | | GraphConv | 2 | $4.0181 \pm 0.13$ | $0.848 \pm 0.11$ | $0.852 \pm 0.10$ |
| | | GAT | 2 | $4.1927 \pm 0.04$ | $0.833 \pm 0.10$ | $0.835 \pm 0.10$ |

Table 14: Results of the experiments on the **UKBB Cardiac** dataset. BL: baselines, $k$: number of neighbours, Transd.: transductive training with only self-loops, GC: graph construction. GNNs out-performing the baselines are underlined, and the best performances of static and dynamic graph constructions are **bold**.

| | Initial edges | Model | $k$ | Test accuracy | Homophily ↑ Train | Homophily ↑ Test | CCNS distance ↓ Train | CCNS distance ↓ Test |
|---|---|---|---|---|---|---|---|---|
| **BL** | - | Majority vote | - | $0.5000 \pm 0.00$ | - | - | - | - |
| | - | Random Forest | - | $0.7027 \pm 0.00$ | - | - | - | - |
| | - | Linear Regression | - | $0.6916 \pm 0.00$ | - | - | - | - |
| **Transd.** | Self-loops | GCN | - | $\mathbf{0.6816 \pm 0.01}$ | $1.000 \pm 0.00$ | $1.000 \pm 0.00$ | $0.000$ | $0.000$ |
| | | SAGE | - | $0.5920 \pm 0.08$ | $1.000 \pm 0.00$ | $1.000 \pm 0.00$ | $0.000$ | $0.000$ |
| | | GraphConv | - | $0.6724 \pm 0.02$ | $1.000 \pm 0.00$ | $1.000 \pm 0.00$ | $0.000$ | $0.000$ |
| | | GAT | - | $0.6812 \pm 0.01$ | $1.000 \pm 0.00$ | $1.000 \pm 0.00$ | $0.000$ | $0.000$ |
| **Static GC** | Random | GCN | - | $0.5019 \pm 0.03$ | $0.504 \pm 0.34$ | $0.480 \pm 0.34$ | $0.500$ | $0.499$ |
| | | SAGE | - | $0.6824 \pm 0.02$ | $0.504 \pm 0.34$ | $0.480 \pm 0.34$ | $0.500$ | $0.499$ |
| | | GraphConv | - | $0.5169 \pm 0.03$ | $0.504 \pm 0.34$ | $0.480 \pm 0.34$ | $0.500$ | $0.499$ |
| | | GAT | - | $0.5291 \pm 0.02$ | $0.504 \pm 0.34$ | $0.480 \pm 0.34$ | $0.500$ | $0.499$ |
| | $k$-NN Euclidean | GCN | 10 | $0.6632 \pm 0.01$ | $0.590 \pm 0.19$ | $0.605 \pm 0.19$ | $0.477$ | $0.467$ |
| | | SAGE | 2 | $0.6498 \pm 0.01$ | $\mathbf{0.778 \pm 0.25}$ | $\mathbf{0.786 \pm 0.25}$ | $\mathbf{0.345}$ | $\mathbf{0.335}$ |
| | | GraphConv | 5 | $0.6686 \pm 0.01$ | $0.640 \pm 0.22$ | $0.645 \pm 0.23$ | $0.449$ | $0.447$ |
| | | GAT | 20 | $0.6322 \pm 0.03$ | $0.563 \pm 0.16$ | $0.572 \pm 0.15$ | $0.488$ | $0.483$ |
| | $k$-NN Cosine | GCN | 20 | $0.6517 \pm 0.00$ | $0.564 \pm 0.16$ | $0.576 \pm 0.15$ | $0.487$ | $0.482$ |
| | | SAGE | 10 | $0.6510 \pm 0.03$ | $0.595 \pm 0.19$ | $0.601 \pm 0.19$ | $0.474$ | $0.470$ |
| | | GraphConv | 10 | $0.6563 \pm 0.02$ | $0.595 \pm 0.19$ | $0.601 \pm 0.19$ | $0.474$ | $0.470$ |
| **Dynamic GC** | No edges | GCN | 2 | $0.6816 \pm 0.01$ | $0.644 \pm 0.36$ | $0.627 \pm 0.37$ | $0.458$ | $0.468$ |
| | | SAGE | 20 | $0.6379 \pm 0.02$ | $0.599 \pm 0.18$ | $0.572 \pm 0.19$ | $0.489$ | $0.484$ |
| | | GraphConv | 2 | $0.6215 \pm 0.06$ | $0.606 \pm 0.37$ | $0.636 \pm 0.36$ | $0.478$ | $0.463$ |
| | | GAT | 2 | $0.6839 \pm 0.00$ | $0.615 \pm 0.38$ | $0.616 \pm 0.38$ | $0.474$ | $0.473$ |
| | Self-loops | GCN | 2 | $0.6521 \pm 0.03$ | $0.622 \pm 0.37$ | $0.602 \pm 0.36$ | $0.470$ | $0.479$ |
| | | SAGE | 20 | $0.6444 \pm 0.01$ | $0.617 \pm 0.20$ | $0.606 \pm 0.21$ | $0.461$ | $0.468$ |
| | | GraphConv | 2 | $0.6659 \pm 0.02$ | $0.718 \pm 0.33$ | $0.652 \pm 0.36$ | $0.405$ | $0.454$ |
| | | GAT | 2 | $0.6812 \pm 0.01$ | $0.636 \pm 0.37$ | $0.634 \pm 0.38$ | $0.463$ | $0.464$ |
| | Random | GCN | 2 | $0.6360 \pm 0.02$ | $0.551 \pm 0.37$ | $0.538 \pm 0.37$ | $0.495$ | $0.497$ |
| | | SAGE | 10 | $0.6678 \pm 0.01$ | $0.508 \pm 0.16$ | $0.556 \pm 0.19$ | $0.500$ | $0.490$ |
| | | GraphConv | 2 | $0.6563 \pm 0.02$ | $0.542 \pm 0.36$ | $0.526 \pm 0.36$ | $0.496$ | $0.499$ |
| | | GAT | 10 | $0.6510 \pm 0.04$ | $0.520 \pm 0.18$ | $0.516 \pm 0.16$ | $0.499$ | $0.499$ |
| | $k$-NN Euclidean | GCN | 2 | $0.6781 \pm 0.01$ | $0.611 \pm 0.37$ | $0.612 \pm 0.36$ | $0.475$ | $0.475$ |
| | | SAGE | 10 | $\mathbf{0.6970 \pm 0.02}$ | $0.499 \pm 0.11$ | $0.507 \pm 0.12$ | $0.500$ | $0.500$ |
| | | GraphConv | 2 | $0.6860 \pm 0.02$ | $0.678 \pm 0.35$ | $0.614 \pm 0.38$ | $0.436$ | $0.474$ |
| | | GAT | 5 | $0.6690 \pm 0.03$ | $0.541 \pm 0.25$ | $0.554 \pm 0.25$ | $0.495$ | $0.493$ |
| | $k$-NN Cosine | GCN | 2 | $0.6770 \pm 0.00$ | $0.607 \pm 0.37$ | $0.595 \pm 0.37$ | $0.477$ | $0.482$ |
| | | SAGE | 10 | $0.6659 \pm 0.03$ | $0.682 \pm 0.24$ | $0.677 \pm 0.25$ | $0.421$ | $0.426$ |
| | | GraphConv | 2 | $0.6862 \pm 0.01$ | $\mathbf{0.767 \pm 0.25}$ | $\mathbf{0.714 \pm 0.30}$ | $\mathbf{0.357}$ | $\mathbf{0.409}$ |
| | | GAT | 2 | $0.6736 \pm 0.01$ | $0.589 \pm 0.37$ | $0.588 \pm 0.37$ | $0.484$ | $0.484$ |

Table 15: Mean and standard deviation of normalised attention values from all neighbours with respective labels of a graph structure with high and low homophily. The highest attention values for each node label class are highlighted in bold. NC: normal control, MCI: mild cognitive impairment, AD: Alzheimer's disease.

| Homophily | Node label | Attention from NC | Attention from MCI | Attention from AD |
|---|---|---|---|---|
| **0.9** | NC | $\mathbf{1.919 \pm 1.08}$ | $0.532 \pm 0.56$ | $0.091 \pm 0.21$ |
| | MCI | $0.198 \pm 0.31$ | $\mathbf{1.881 \pm 1.06}$ | $0.083 \pm 0.22$ |
| | AD | $0.158 \pm 0.29$ | $0.777 \pm 0.66$ | $\mathbf{1.961 \pm 1.14}$ |
| **0.4** | NC | $0.978 \pm 0.75$ | $\mathbf{2.002 \pm 1.05}$ | $0.255 \pm 0.34$ |
| | MCI | $0.556 \pm 0.59$ | $\mathbf{0.972 \pm 0.74}$ | $0.243 \pm 0.36$ |
| | AD | $0.676 \pm 0.68$ | $\mathbf{1.743 \pm 0.97}$ | $0.940 \pm 0.71$ |
| **0.1** | NC | $0.000 \pm 0.00$ | $\mathbf{3.106 \pm 1.47}$ | $0.415 \pm 0.48$ |
| | MCI | $\mathbf{0.985 \pm 0.74}$ | $0.000 \pm 0.00$ | $0.461 \pm 0.57$ |
| | AD | $1.038 \pm 0.88$ | $\mathbf{3.013 \pm 1.35}$ | $0.000 \pm 0.00$ |

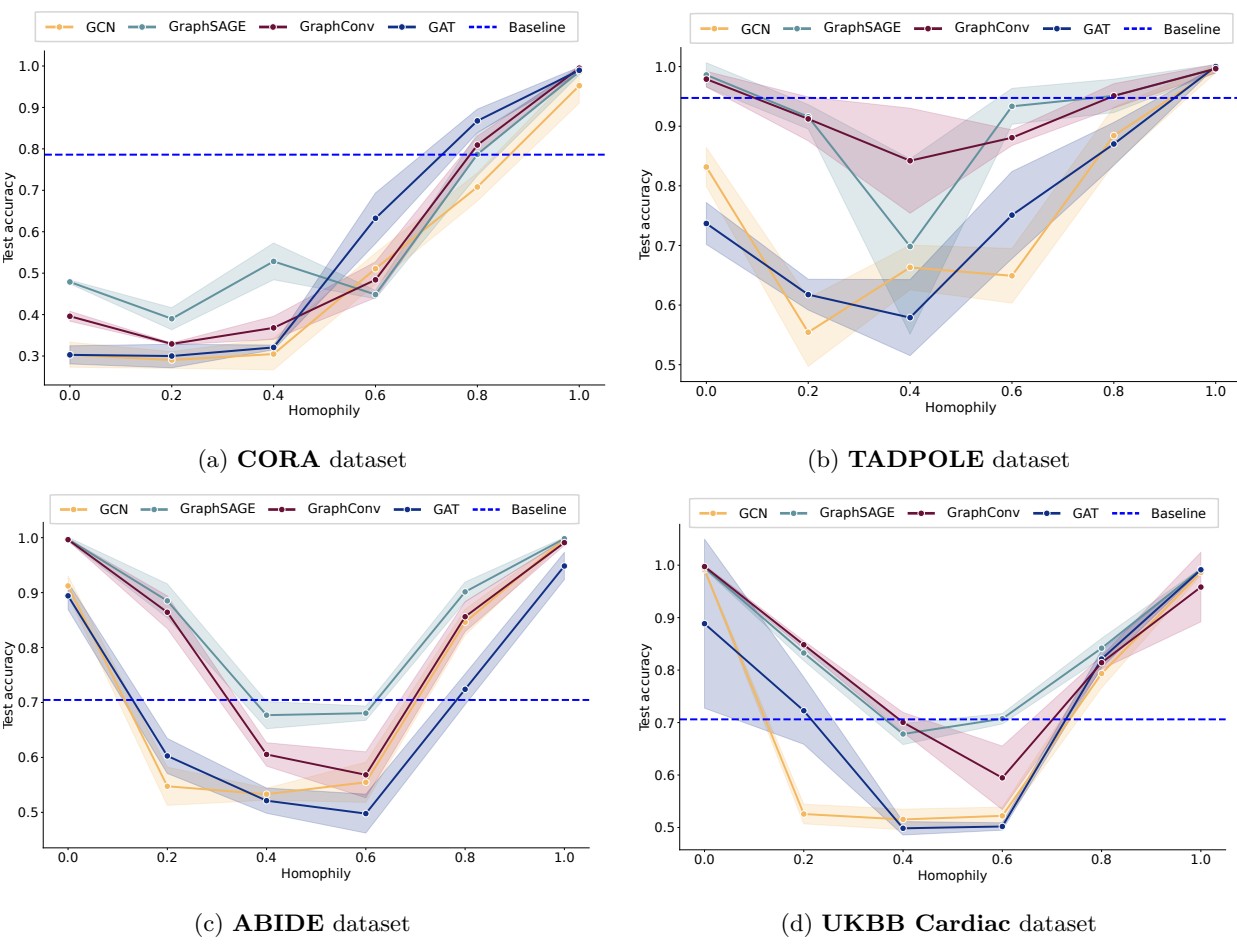

(a) **CORA** dataset

(b) **TADPOLE** dataset

(c) **ABIDE** dataset

(d) **UKBB Cardiac** dataset

Figure 6: Performance of different graph convolutions on synthetic graph structures with different homophily values on (a) the **CORA** dataset, (b) the **TADPOLE** dataset, (c) the ABIDE dataset, and (d) the UKBB cardiac dataset. The dashed blue line indicates the mean performance of the best baseline for each dataset.

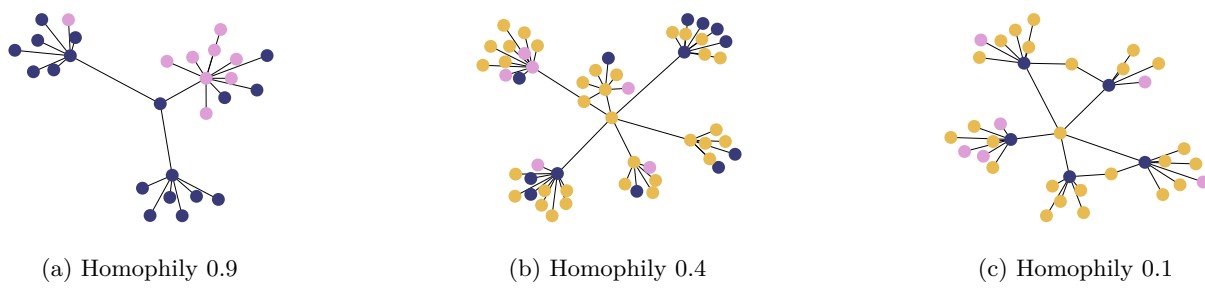

(a) Homophily 0.9

(b) Homophily 0.4

(c) Homophily 0.1

Figure 7: **Visualisation of attention-based neighbourhoods** of a random node (centre node) from the TADPOLE dataset with synthetically generated graph structures and its two-hop neighbourhood. The node colours indicate node labels and the distance is proportional to the summed attention weight of the edges to the respective neighbouring node.

