# OpenReview forum: "Are Population Graphs Really as Powerful as Believed?"
_TMLR — Accepted by TMLR_

### Review · Reviewer_amZq · 2023-12-08

**Summary Of Contributions:**

This work is an empirical benchmarking study that assesses the predictive performance of graph convolutional networks (GCNs) applied to data with population graph structure (encoding relationship among data instances, represented as nodes), compared to baselines. The central finding of the work is that, in nearly all cases, well-tuned baselines (logistic/linear regression, ridge regression, and random forests), are competitive with or outperform GCNs on population graph data. Notably, these baseline results are higher than reported elsewhere in the literature. In cases where GCNs perform well, the population graph is given and carries meaningful information not accounted for by the node features (here, empirically in experiments with the CORA and Pubmed datasets). GCNs perform relatively less well in settings where the population graph structure is inferred from the data. They further conduct experiments with multi-modal data, and find that learned population graphs and GCNs do not generally improve predictive performance if all data modalities are incorporated into the node features and used as predictors. They further show that the performance of GCNs is highly dependent on the graph structure learned with a dataset-dependent relationship between performance of GCNs and homophily.

**Audience:**

Yes

**Claims And Evidence:**

Yes

**Requested Changes:**

Critical
  * Revise the manuscript to more clearly substantiate the conclusion that the bottleneck in performance is the graph construction method, or adjust the conclusion if it is not appropriate.


Minor changes [Not critical]
  * Fully define the set of symbols used in equations (5), (6), (7), (8)
  * The argument in section 5.5 around image-derived features posing a challenge seems to be framed in an overly strong way given that it is claimed that are the “only exception” to the idea of using the full set of node features to learn the edges. Shouldn’t it be the case that this applies to all high-dimensional or complex data (e.g. text, audio)? Furthermore, it doesn’t seem like this poses any fundamental problems other than computational ones. Please revise for clarity here and also in section 5, point 2.
  * Please revise: In section 5.3 there are two sentences that seem to be redundant and somewhat contradictory. “The method of generating a k-NN graph, which is most frequently used for population graphs performs on pair with the baseline and often even under-performs the baseline. Using a random graph structure or a k-NN graph negatively impacts model performance, both in the static and dynamic cases”. “pair” is also a typo.
  * There is an opportunity to include more convincing examples in the conclusion for which population graph structure might be useful. A naive example might be public health analyses that include geospatial aspects (i.e., modeling effect of zip codes or neighborhood on health outcomes).

**Strengths And Weaknesses:**

Strengths
  * This is all-together a compelling empirical study that is well-motivated and reasonably comprehensive.
  * The study has high potential for impact on future research into the use of learned population graphs and GCNs. This is arguably a niche domain, but I believe that the results will be of interest to that community.
  * The paper is generally well written. I particularly appreciate the detailed presentation of the background material, related work, and presentation of the experiments and results.

Weaknesses
  * There are a few places where the clarity of the manuscript could be improved, particularly with regards more tightly connecting the hypotheses and evidence generated to the conclusions about the need for better graph construction methods. To be clear, I believe that the elements needed to make these more clear are there, but they could be better presented. For examples, the experiments do support the conclusion that there is a need to identify ways to incorporate structure and knowledge into the graph that is not accounted for by the node features, but this seems to me to be categorically different from the claim that we need to improve the methods for constructing a graph from a set of node features. However, in the conclusion section, these two things are not clearly distinguished from one another. Furthermore, it seems that there actually is some evidence from Figure 4 that population graph learning could be beneficial if a graph with the appropriate level of homophily for a given dataset and problem could be learned, which ultimately supports the work’s conclusion, but is not clearly presented as evidence for that argument.

---

> ### Author Response · Authors · 2024-01-11
> **Response to Review**
>
> We thank the reviewer for their insights and remarks and for engaging with our research. Based on their suggestions, we revised to clarify the conclusion and made the following changes to our manuscript:
>
> We added the following bullet point to Section 6:
> “Better graph construction methods are required. The experiments on the benchmark datasets and the synthetically generated graph structures with different homophily values (see Figures 4 and 5), show that GNNs can improve downstream task performance if the graph structure is “meaningful”. However, current graph construction methods do not lead to valuable graph structures, which makes graph construction the performance bottleneck in these settings. The same is represented by the fact that random graph structures often achieved comparable results to approaches like k-NN graphs.”
>
> And the following paragraph to Section 7:
> “We see a need for either more advanced methods to learn a graph structure that contains additional meaningful information to the node features or novel ideas on how to build population graphs from additional information that cannot be represented as node features. When using synthetically generated graph structures, we observe that only graphs with higher homophily than possible to extract from the node features result in better performance of GNNs compared to properly tuned graph-agnostic methods such as a random forest or linear regression (Figure 4).“
>
> Furthermore, we added the following sentence to the abstract, highlighting the need for either better graph construction methods or different applications for population graphs:
> “Based on our results, we argue towards the need for “better” graph construction methods or innovative applications for population graphs to render them beneficial”
>
> In addition, we re-structured large parts of Section 5, moved several experiments to the appendix and disentangled the experiments on benchmark datasets (e.g. CORA) and population graph datasets completely. Furthermore, we added a stronger introduction to population graphs, the medical motivation for using them and how they have been used in related works to Sections 1 and 3. We believe that these changes improve readability and the thread of the argument in the paper.
>
> We also thank the reviewer for highlighting additional minor points of change and made the following adjustments:
> 1. We included definitions for all symbols.
>
> 2. We changed the second bullet point in Section 6 in the following way:
> “The utilisation of population graphs with the goal of multi-modal data integration might not be as promising as believed. The most frequently used method for the construction of population graphs includes a separation of features into node features, and ones utilised for edge construction. We show that using all available features for edge construction and as node features might lead to better results and argue that a concatenation of all features is easily doable --except when using high dimensional data as node features. We see potential in using population graphs in different settings, where the connectivity information cannot easily be integrated with the node features.”
>
> We added the following clarification to Section 5.3 to support our argument:
> “In this so far typical setting of population graphs, we scrutinise this claimed advantage and argue that all available features can easily be appended and therefore incorporated into the node features. However, we see exceptions, such as when the information used for edge construction cannot be used as node features. This is e.g. the case when the information used for edge construction cannot be used as node features. This is the case when high dimensional data is used as node features --e.g. text, audio data or images. However, this setup comes with large memory requirements and has not been studied in detail. We encourage a more critical assessment of the utilisation of GNNs for multi-modal data integration in conventional configurations of population graphs and advocate a progression towards more advanced settings and a more suitable usage of multi-modal data integration for cases where it is indeed necessary.”
>
> 3. We thank the reviewer for their remark and changed the here addressed formulation. Since we restructured parts of Section 5, this content is now in the appendix:
> “With the experiments on the CORA dataset (Table 11), we observe that only the GNNs that utilise the “ground truth” edges out-perform our baseline methods, while the commonly used graph construction methods for population graphs also do not benefit performance on the CORA dataset. We observed similar results for the other benchmark datasets.”
>
> THIS COMMENT IS CONTINUED BELOW

---

> ### Author Response · Authors · 2024-01-11
> **Response to Review - continued**
>
> CONTINUED FROM ABOVE
>
> 4. We added more examples to the conclusion for potential promising directions of future work for population graph studies:
>
> “We generally see three future directions for population graph studies. Either (a) new and better graph construction methods need to be developed for population graphs to bring benefits to medical downstream tasks, (b) innovative applications of population graphs that truly benefit from the usage of connectivity information need to be explored, or (c) the usage of population graphs in combination with GNNs does not seem valuable for the performance of medical downstream tasks. For better graph construction methods, we see the requirement of increasing the information content of the graph structure compared to the node features alone. This could potentially be achieved by encoding information in the graph structure that cannot be trivially added to the node features, such as genetic similarity between subjects or the risk groups in survival analysis. Other potentially interesting applications of population graphs, where the edges add additional information to the node features, are geospatial graphs. This could include analysing location-based health data, disease spreading, or tracing local differences in medication or care units. Also, time-series data has not been explored in great detail in the context of population graphs, which might add more valuable information to the graph structure. The concept of population graphs has, with a few exceptions (Keicher et al., 2021), mostly focused on vector data instead of images. This is mostly because image data is much larger and, therefore, more difficult to fit into memory in the context of population graphs. This could be another interesting future direction to improve the predictive power of population graphs.”

---

> > ### Comment · Reviewer_amZq · 2024-01-28
> >
> > Thank you for the detailed revisions and response. You have addressed my concerns.

---

### Review · Reviewer_PnUe · 2023-12-13

**Summary Of Contributions:**

The authors of the submission investigated the question whether the population graphs can help consequent data analysis in biomedical applications, comparing to the baselines without using population graphs.

The presented work is essentially a benchmarking study on how GNNs can be constructed based on population graphs in different settings
and comparing different graph learning methods to baseline models. The authors showed the comparison results on both citation network datasets (CORA, CITESEER, PUBMED) and five medical population graph datasets. Different graph construction methods, including static graph construction methods and dynamic graph construction methods, and different GNNs and learning methods including neural sheaf diffusion were tested to show the performance.

The conclusion is that without good population graph construction, graph learning methods may not perform better than graph-agnostic methods.

**Audience:**

Yes

**Claims And Evidence:**

No

**Requested Changes:**

See above.

**Strengths And Weaknesses:**

Strengths:

1. It is an important question to ask whether the concept of population graphs is meaningful.
2. Different graph construction and learning methods have been tested to show empirical results.

Weaknesses:

1. There is no new method development, which may make the submission not suitable for the journal.
2. There is no theoretical analysis on why population graph is or is not useful. Empirical results on limited setups and benchmark datasets can provide some evidence but may not be generalizable.
3. The authors may want to give some insights why population graphs might be helpful at the beginning. Was that based on 'guilt association' assumption? If so, some discussion or analysis should be provided on what consequent tasks the populations graphs can or can not help.
4. The authors claimed to test neural sheaf diffusion models in their experiments, but only the results on citation networks were provided but there were no results on medical datasets.
5. The presentation can be improved. For example, when explaining the results shown in Figure 4. The figure caption stated that the results are based on synthetic graph construction while in the text, it was stated that 'Figure 4 shows the performance of different graph convolutions on 3-layer GNNs using static graph construction on the different datasets.' The authors need to be clear on how graph construction was done to achieve different homophily levels.
5. The presented results should be better organized to help understand when population graphs can be helpful.

---

> ### Author Response · Authors · 2023-12-21
>
> We thank the reviewer for their comments. Could we kindly ask the reviewer to clarify which part of the experiments or datasets they mean when they note a 'limited setup' in point 2. This would greatly help us to address this comment appropriately.

---

> > ### Comment · Reviewer_PnUe · 2024-01-05
> >
> > The main concern is the lack of theoretic analysis, which might not provide generalizability of the conclusions based on limited experimental results.
> >
> > For example, many different GNN architectures as well as message passing schema have not been tested, including many high-order message passing schema. When generating random graphs, different families of random graphs may lead to different conclusions.

---

> > > ### Author Response · Authors · 2024-01-11
> > > **Respone to Comment**
> > >
> > > We appreciate the reviewer’s insights and refer to the response to the initial comment regarding the theoretical analysis. While we only use one method for random graph construction, we argue that the purpose of those experiments is to show that random graphs are able to achieve similar performance to graphs constructed with “targeted” graph construction (k-NN), hence it is less important if other random graph construction methods yield better or worse performance. We did observe these results with Erdos Renyi graphs over all datasets and believe that additional random graph construction methods would not lead to additional or different conclusions. We note that for every dataset and training setup, simply by hyperparameter tuning, it was possible to generate a random graph that led to good performance. We therefore believe that our conclusion holds, even if not every individual randomly generated graph achieves good performance.
> > >
> > > We find the idea of using higher-order GNNs an interesting possible next step. However, many higher-order approaches, such as [1,2,3], are only useful for graph-level or sub-graph-level predictions. Other approaches, such as [4,5], that discuss the usage of higher-order GNNs for node-level predictions use temporal/sequential data such as click-stream data, where the choice of a next page is dependent on several prior data points. We would see this as a fitting method for sequential medical data, i.e. longitudinal studies. However, this is not the use case in our experiments, where no temporal data is available. Furthermore, we note that the similarity between subjects in a population graph decreases quickly with higher-order structures of the graph. We are, therefore, not convinced that they would benefit this specific use case. We would be interested in hearing the reviewer’s thoughts on this. We deem the evaluation of higher-order GNNs for population graph studies interesting next steps to investigate in future work and added this to Section 7:
> > >
> > > “Furthermore, investigating other graph convolutions or different GNN architectures in combination with specific population graph setups might give more insights. One example would be higher-order GNNs for node-level predictions (Li et al., 2021). We would see this as a fitting method for longitudinal studies.”
> > >
> > > We believe that we tested the most widely used and suitable architectures in this context. If the reviewer thinks that the inclusion of a specific important GNN architecture would lead to substantially different conclusions, we would be grateful for them to point us towards this work.
> > >
> > > [1] Morris, Christopher, et al. "Weisfeiler and Leman go neural: Higher-order graph neural networks." Proceedings of the AAAI conference on artificial intelligence. Vol. 33. No. 01. 2019.
> > >
> > > [2] Gao, Jianliang, et al. "Higher-order interaction goes neural: A substructure assembling graph attention network for graph classification." IEEE Transactions on Knowledge and Data Engineering (2021).
> > >
> > > [3] Xiong, Ping, et al. "Efficient Computation of Higher-Order Subgraph Attribution via Message Passing." International Conference on Machine Learning. PMLR, 2022.
> > >
> > > [4] Yuan, Kaixin, Jing Liu, and Jian Lou. "Higher-Order Masked Graph Neural Networks for Traffic Flow Prediction." 2022 IEEE International Conference on Data Mining (ICDM). IEEE, 2022.
> > >
> > > [5] Jin, Di, et al. "Graph neural network for higher-order dependency networks." Proceedings of the ACM Web Conference 2022. 2022.

---

> > > > ### Comment · Reviewer_PnUe · 2024-01-30
> > > >
> > > > The authors did perform reasonable empirical study to understand how population graph may help consequent downstream tasks and the good population graph construction is needed to achieve that. Their revision is more organized and with clarified conclusion.
> > > >
> > > > It might be my bias that it would be better if the authors can provide deeper theoretical/quantitative analysis on how the error/uncertainty in constructed population graph propagate to downstream predictions. Otherwise, the conclusion is more or less general 'garbage in, garbage out' statement, which may have limited significance.

---

> ### Author Response · Authors · 2024-01-11
> **Response to Review**
>
> We thank the reviewer for their insights and advice, on which we made several changes to our manuscript. We here follow the numbering from the original review in addressing their comments:
>
> 1. We acknowledge the reviewer’s remark regarding the technical novelty of our paper. We would like to highlight that while methodological novelty is indeed one aspect of innovation, novelty can be seen in various forms, including the here highlighted gap in research, which leads to us questioning the utility of population graphs the way they are currently used. We would like to remark that methodological novelty is not a requirement for acceptance at TMLR but rather an interest to the community.
>
> 2. We would like to highlight that the primary aim of our paper is to conduct an empirical analysis in order to inform which areas of population graph research require further theoretical analysis. Based on our experiments, we identified the graph construction as the performance bottleneck for population graphs and see potential future analyses in this direction on what an ideal or beneficial graph construction method would need to satisfy. This could e.g. include analyses of a theoretical upper bound of the information content captured by the graph structure –given its construction method. We agree with the reviewer that a theoretical analysis of the expressiveness and power of population graphs would be interesting and important and we believe that the work presented here informs this the direction for further theoretical analyses. We have added this to the future work section in our manuscript:
>
> “We generally see three future directions for population graph studies. Either (a) new and better graph construction methods need to be developed for population graphs to bring benefits to medical downstream tasks, (b) innovative applications of population graphs that truly benefit from the usage of connectivity information need to be explored, or (c) the usage of population graphs in combination with GNNs does not seem valuable for the performance of medical downstream tasks. It would be interesting to follow up with a theoretical analysis. Our experiments showed that the graph construction is a major performance bottleneck for population graphs. We believe this to be a good starting point for follow-up analyses. For better graph construction methods, we see the requirement of increasing the information content of the graph structure compared to the node features alone.”
>
> 3. The initial assumption for using population graphs is that neighbouring subjects are similar, and therefore, the graph learning benefits from information exchange between them. This follows the medical motivation that subjects with similar phenotypes share similar pathologies. The hope is to improve personalised medicine with this approach. One could compare this to a “reasoning by precedence”. We added the following paragraph to Section 1:
> “So far, there are two commonly used arguments for using medical population graphs compared to graph-agnostic models: (1) GNNs allow for meaningful multi-modal data integration, and (2) the message passing across neighbourhoods improves model performance. The idea behind the latter is that similar subjects will be neighbours in the graph and therefore share similar medical information and most likely also require similar predictions. This is based on the assumption that the learning pipeline benefits from the sharing of information across neighbouring subjects in the population graph if they are similar to each other.”
>
> 4. The reviewer might have overlooked this. We kindly refer to Table 2, where we report the results of neural sheaf networks for all classification datasets, including the medical datasets. More details about hyperparameters and learning setups can be found in Appendix C Table 10.
>
> 5. We thank the reviewer for their remark and changed the text on Figure 4 in the following way:
> “In this section, we investigate the impact of the graph structure on model performance from three further viewpoints. (1) The experiments above have indicated that the graph structure has a different impact on different graph convolutions, (2) the complexity of the dataset plays an important role in the performance of GNNs on low-homophily graphs, and (3) if a meaningful graph structure is available, GNNs out-perform graph-agnostic models. Therefore, we perform additional experiments on synthetically generated graph structures at different homophily values. Here, the graph is constructed statically and to specifically match a certain homophily value. The results for three datasets are visualised in Figure 4, and more visualisations can be found in the appendix Figure 7.”
>
> Furthermore, we adapt the part where we refer to the experiments in Section 5.4 as follows:
> “Figure 4 shows the performance of different graph convolutions on 3-layer GNNs using synthetically generated graph structures for the different datasets.”
>
> CONTINUED BELOW

---

> ### Author Response · Authors · 2024-01-11
> **Response to Review - continued**
>
> CONTINUED FROM ABOVE
>
> 6. We changed the following paragraph in the conclusion to highlight the potential benefits of population graphs and future research directions in this area:
>
> “Our experiments showed that the graph construction is a major performance bottleneck for population graphs. We believe this to be a good starting point for follow-up analyses. For better graph construction methods, we see the requirement of increasing the information content of the graph structure compared to the node features alone. This could potentially be achieved by encoding information in the graph structure that cannot be trivially added to the node features, such as genetic similarity between subjects or the risk groups in survival analysis. Other potentially interesting applications of population graphs, where the edges add additional information to the node features, are geospatial graphs. This could include analysing location-based health data, disease spreading, or tracing local differences in medication or care units. Also, time-series data has not been explored in great detail in the context of population graphs, which might add more valuable information to the graph structure. The concept of population graphs has, with a few exceptions (Keicher et al., 2021), mostly focused on vector data instead of images. This is mostly because image data is much larger and, therefore, more difficult to fit into memory in the context of population graphs. This could be another interesting future direction to improve the predictive power of population graphs.”
>
> Furthermore, we moved a portion of the results to the appendix to improve clarity and readability and ended the abstract with a specific note on what we see necessary for population graphs to be beneficial:
> “Based on our results, we argue towards the need for “better” graph construction methods or innovative applications for population graphs to render them beneficial.”
>
> We also add the following paragraph at the end of Section 1:
> “Our results lead us to conclude that we need a discussion about whether population graphs are beneficial over graph-agnostic methods and that the currently available graph construction methods are the performance bottleneck of GNNs on population graphs. We see a requirement for better graph construction methods if we want to improve the performance of GNNs on population graphs.”
> Furthermore, we restructured parts of the main text. We e.g. moved the experiments of attention evaluation to the appendix, disentangled all benchmark experiments from the medical population graph experiments, and merged the prior sections 5.1 and 5.2. Furthermore, we shortened the section on the dataset introduction and moved parts of it to the appendix. With these changes, we believe to make the argument easier to follow and more strongly emphasise our findings that population graphs do not achieve higher performance on all tested medical datasets compared to well-tuned baseline methods.

---

### Review · Reviewer_ZoVr · 2023-12-19

**Summary Of Contributions:**

This paper provides a comprehensive study investigating the use of graph neural networks (GNN) for population graph-aided machine learning in clinical healthcare applications. There is further study of appropriate usage of GNN in general using three benchmark datasets. Ultimately, the paper's strongest contribution is the thorough manner in which it investigates prior claims surrounding the benefit of population graphs in predictive modeling among medical domains.

**Audience:**

Yes

**Broader Impact Concerns:**

None that I can ascertain.

**Claims And Evidence:**

Yes

**Requested Changes:**

A revision of Section 1 to provide a more distinct introduction to population graphs and how they have been used. This could also be done by reformatting Section 3 to more clearly describe and discuss prior approaches using GNNs with medical data.

Relegating the focus on common benchmark datasets to justify the algorithmic maturity of the experiments to the Appendix or Supplement. This would be to tighten the paper's focus to be solely about population graph based approaches. I am not dead-set on this being a requirement for my support of the paper but I do feel that it would greatly enhance the impression the paper leaves. I'd be interested to hear the authors' perspective on this point and whether it is critical to include such an exhaustive set of discussion points around these when they don't contribute to the claims made in the paper.

An additional experiment investigating the effect of dataset size. If it is not possible to use a new dataset such as MIMIC, I suppose that this scaling experiment could be used by partitioning the UKBB Brain Age dataset and varying over the total number of nodes and analyzing algorithm performance?

**Strengths And Weaknesses:**

**STRENGTHS**

As mentioned above, the paper takes a lot of care to thoroughly and rigorously evaluate the performance of GNN based prediction models and graph-agnostic baseline approaches for a variety of medical datasets becoming increasingly common within the GNN literature. The paper calls into question the benefit of these GNN methods in a systematic fashion over multiple axes of inquiry with regard to the formulation and evaluation of GNN algorithmic techniques.

I appreciated the care and attention taken by the authors to provide sufficient background information in Section 2 which made the rest of the paper far more accessible to a non-expert such as myself. I felt prepared to follow the insights developed through the experiments, validating the claims made at the end of Section 1. The claims and their justification were clearly articulated throughout the paper.

**WEAKNESSES**

However, I do feel that the messaging and focus of the paper was a little inconsistent at times. I believe that a much firmer focus on population graphs and their intended use needs to be made at the outset of the paper. It felt that the description/definition of population graphs in Section 1 was a bit hurried. This made the initial discussion in Section 4 (and the related work in Section 3) that focused on GNNs in general seem to be a bit of a deviation from the primary direction the paper proposes at the beginning. There's a little too much effort to describe and validate results on the benchmark datasets in my opinion as they have fairly little in common with the insights that are intended to be the primary takeaway from this paper.

Perhaps the attention paid on these benchmarks is to satisfy the GNN community and demonstrate that the authors have done their due diligence in developing + replicating GNN algorithm performance? If this is the case, I'd highly recommend relegating the discussion about the benchmark datasets (and the associated analyses) to a supplementary material with clear references to them in the paper. This would serve to enhance the impression that I believe the authors are intending, to demonstrate that the benefit of population graphs for prediction in medical settings is not yet clear and is perhaps unwarranted.

One question that I also feel is unaddressed by the analyses in this paper is the question of scale. The five medical population graph datasets used are quite small in comparison to contemporary approaches in ML today (and among other publically available Medical databases such as MIMIC, eICU, and HiRID). I am left wondering if it is possible that these population graph focused use-cases require another order of magnitude (or several) in dataset scale and variety? The inclusion of an analysis that looked at this question of dataset size (and perhaps using an additional medical benchmark dataset) would definitely satisfy my concerns and allow me to strongly advocate for this paper's acceptance for publication

---

> ### Author Response · Authors · 2024-01-11
> **Response to Review**
>
> We thank the reviewer for their constructive comments and advice and for engaging with our research. Based on their input, we made the following changes to our manuscript:
>
> 1. We added the following paragraph to Section 1, extending the introduction to population graphs:
> “Several works have shown that population graphs for medical applications can improve downstream tasks compared to graph-agnostic methods (Parisot et al., 2017; Kazi et al., 2019; Cosmo et al., 2020; Kazi et al., 2022). Parisot et al. (2017) first introduced the concept of population graphs for the detection of Alzheimer’s disease and autism. Later works (Kazi et al., 2019; Cosmo et al., 2020; Kazi et al., 2022; Bintsi et al., 2023a;b) used the method of population graphs under different settings, developing new graph construction methods and for different tasks, such as age prediction. The motivation for using population graphs is the hypothesis that subjects that share similar phenotypes, tend to have similar pathologies and therefore benefit from sharing information. The goal is to facilitate personalised medicine by utilising the shared information across similar subjects.”
>
> Furthermore, we restructured Section 3 to more clearly recap the usage of population graphs and how they have been used. Concretely, we moved the paragraph about graph assessment methods to the end of Section 3 to refocus the section on population graphs instead, extended the usage of GNNs and population graphs in related works in Section 3.
>
> 2. We agree with the reviewer that the experiments on the benchmark datasets are not the core experiments of this study. However, we argue that the results are an important part of showing that a meaningful graph structure that adds additional information to the node features (which we assume for the citation networks), leads to improved performance of GNNs in comparison to graph-agnostic methods. Yet, we agree that the manuscript benefits from a disentanglement of these two datasets and that the focus should clearly be on population graphs. We therefore made the following changes to the manuscript regarding the experiments on the benchmark citation datasets:
> (a) We removed the overview of the benchmark citation datasets from Tables 1 and 2 and created a separate Table 9 for these results.
> (b) We completely removed the baseline evaluation from Table 5.
> (c) We separated the experiments on the population graph datasets from the benchmark citation networks and moved the former to Section 5.4 on “Ground Truth Graph Structures”.
> (d) We discuss the experiments on the citation network datasets solely in the context that a meaningful graph structure indeed helps the performance of GNNs, supporting our hypothesis that GNNs with current graph construction methods are also not beneficial. More details about the benchmark datasets were moved to the appendix.
>
> 3. We thank the reviewer for their suggestion and added additional experiments regarding the scale of the dataset to Section 5.4. We (1) –as suggested by the reviewer– partition the largest population graph dataset (UKBB brain age) into smaller subsets of 25%, 50% and 75% of the original dataset and compare GNN performance to baseline performance. Additionally, we generate a synthetic dataset with 5000, 10000, 20000, and 30000 nodes to evaluate the impact of much larger graphs. In all these experiments, we observe the same pattern that GNNs do not show an improvement in downstream performance compared to well-tuned baselines.

---

> > ### Comment · Reviewer_ZoVr · 2024-01-12
> > **Thanks!**
> >
> > Thank you authors. I've had the chance to look back through the revised paper and look through your comments. I appreciate the attention you've paid to the recommendations and questions that I and the other reviewers have provided. As a result the paper is much improved.

---

### Decision · Action_Editor_jF6r · 2024-02-06

**Recommendation:** Accept as is

**Comment:**

The reviewers found the empirical study compelling, although one reviewer would have preferred accompanying theoretical insights. While I believe that this would have strenghtened the paper, the current findings are sound and would be of interest to the audience.

**Audience:**

Despite the lack of a technical analysis, reviewers believed the results of this paper would be of interest to the community.

**Claims And Evidence:**

This paper provides an interesting empirical study on the use of population graphs.

The authors have worked on the revision extensively and satisfied most of the concerns. The claims and analyses are sound.